# OPTIMALITY OF MATRIX MECHANISM ON $\ell_p^p$-METRIC

**Zongrui Zou, Jingcheng Liu**[*]
State Key Laboratory for Novel Software Technology, New Cornerstone Science Laboratory
Nanjing University
163 Xianlin Avenue, Nanjing, Jiangsu Province, 210023, China
`zou.zongrui@smail.nju.edu.cn, liu@nju.edu.cn`

**Jalaj Upadhyay**[†]
Management Science & Information Systems Department
Rutgers University
100 Rockafeller Road, Piscataway, NJ 08854, USA
`jalaj.upadhyay@rutgers.edu`

## ABSTRACT

In this paper, we introduce the $\ell_p^p$-error metric (for $p \geq 2$) when answering linear queries under the constraint of differential privacy. We characterize such an error under $(\varepsilon, \delta)$-differential privacy in the natural add/remove model. Before this paper, tight characterization in the hardness of privately answering linear queries was known under $\ell_2^2$-error metric (Edmonds et al. (2020)) and $\ell_p^2$-error metric for unbiased mechanisms in the substitution model (Nikolov & Tang (2024)). As a direct consequence of our results, we give tight bounds on answering prefix sum and parity queries under differential privacy for all constant $p$ in terms of the $\ell_p^p$ error, generalizing the bounds in Henzinger et al. (2023) for $p = 2$.

## 1 INTRODUCTION

Analysis or learning with sensitive datasets under privacy has garnered increasing attention in recent years. In this paper, we study the most fundamental question of answering linear queries on confidential dataset $x \in \mathbb{R}^n$ while preserving *differential privacy* (DP) (Dwork et al., 2016). Informally speaking, differential privacy captures the property of a randomized algorithm that its output distribution is relatively stable when executed on two *neighboring datasets*, i.e., datasets that can be formed by changing one data point. More formally,

**Definition 1.1** *Let $\mathcal{M} : \mathcal{X} \to \mathcal{R}$ be a randomized algorithm, where $\mathcal{R}$ is the output domain. For fixed $\varepsilon > 0$ and $\delta \in [0, 1)$, we say that $\mathcal{M}$ preserves $(\varepsilon, \delta)$-differential privacy if, for any measurable set $S \subseteq \mathcal{R}$ and any pair of neighboring datasets $x, y \in \mathcal{X}$, $\mathbf{Pr}[\mathcal{M}(x) \in S] \leq \mathbf{Pr}[\mathcal{M}(y) \in S] \cdot e^\varepsilon + \delta$. If $\delta = 0$, we say $\mathcal{A}$ preserves pure differential privacy (denoted by $\varepsilon$-DP).*

Many fundamental analyses can be cast as a set of linear queries (Dwork & Roth, 2014; Vadhan, 2017): given an input $x \in \mathbb{R}^n$, a set of $m$ linear queries can be represented as the rows of a matrix $A \in \mathbb{R}^{m \times n}$. The answer to the set of queries is simply the matrix-vector product $Ax$. Here, $x, x' \in \mathbb{R}^n$ are neighboring if $\|x - x'\|_1 \leq 1$ (known as *add/remove model* of privacy). When these queries are answered using a privacy-preserving algorithm, $\mathcal{M}$, the performance of the algorithm is usually measured in terms of its *absolute error* or *mean squared error* (eq. (13)).

---

[*]JL and ZZ have been supported by National Science Foundation of China under Grant No. 62472212 and the New Cornerstone Science Foundation.

[†]JU's research was funded by the Rutgers Decanal Grant no. 302918 and an unrestricted gift from Google.

In this paper, we initiate the study of $\ell_p^p$-error metric that seamlessly interpolate[1] between $p = 2$ (squared error) to $p = \infty$ (absolute error):

$$\text{err}_{\ell_p^p}(\mathcal{M}, A) := \max_{x \in \mathbb{R}^n} \left( \mathbb{E} \left[ \|\mathcal{M}(x) - Ax\|_p^p \right] \right)^{1/p}. \tag{1}$$

The error metric defined above has a natural and intuitive interpretation for data analysis. To elaborate on this, consider the most natural mechanism that adds i.i.d. noise to each answer of a set of linear queries, and let $v_i$ be the error in answering the $i$-th query. Then our error metric captures the $p$-th moment of the error, which is a random vector $v \in \mathbb{R}^m$. By considering all $p$, one can identify the exact nature of the probability distribution of the error.

One popular mechanism for privately answering linear queries under different error metrics is the *matrix mechanism* (Li et al., 2015), also known as the *factorization mechanism*. In the matrix mechanism, given a set of $m$ linear queries represented by a *workload matrix* $A \in \mathbb{R}^{m \times n}$, we compute a factorization $LR = A$ (where $L \in \mathbb{R}^{m \times k}$, $R \in \mathbb{R}^{k \times n}$) and output $L(Rx + z)$ for any input $x \in \mathbb{R}^n$ with an appropriately scaled Gaussian random vector $z \in \mathbb{R}^k$. This mechanism is both *unbiased* (i.e., $\mathbb{E}[z] = \mathbf{0}$) and *oblivious*, i.e., the distribution of $z$ is stochastically independent of $x$. In this paper, we show that the optimal matrix mechanism is also optimal among all differentially private mechanisms with respect to the $\ell_p^p$ metric, up to logarithmic factors:

**Theorem 1.2 (Informal statement of Theorem 1.4 and Theorem D.1)** *Fix $A \in \mathbb{R}^{m \times n}$ be a matrix representing $m$ linear queries, and let $\mathcal{M} : \mathbb{R}^n \to \mathbb{R}^m$ be any $(\varepsilon, \delta)$-DP algorithm. Then, there exists a factorization of $A = LR$ such that $\mathcal{M}_{\text{matrix}}(x) = L(Rx + z)$ with $z \sim \mathcal{N}(\mathbf{0}, \|R\|_{1 \to 2}^2 \mathbb{I}_k)$ preserves $(\varepsilon, \delta)$-DP and that $\text{err}_{\ell_p^p}(\mathcal{M}_{\text{matrix}}, A) \lesssim \text{err}_{\ell_p^p}(\mathcal{M}, A) \cdot \text{polylog}(1/\delta, m)$. Here, $\mathbb{I}_k \in \mathbb{R}^{k \times k}$ is the identity matrix.*

To prove this, we characterize the $\ell_p^p$-error for answering linear queries under $(\varepsilon, \delta)$-differential privacy generalizing Edmonds et al. (2020), and also obtain a characterization of $\text{err}_{\ell_p^p}(\mathcal{M}_{\text{matrix}}, A)$ that is tight up to log factors, for every query matrix $A$ and $p \geq 2$. For the convenience of use, we start by stating a weaker form of our lower bound. We will see that this is an immediate corollary of our main theorem.

**Theorem 1.3** *Let $A \in \mathbb{R}^{m \times n}$ be a matrix representing $m$ linear queries. Then for any $(\varepsilon, \delta)$-DP algorithm $\mathcal{M}$, $\text{err}_{\ell_p^p}(\mathcal{M}, A) = \Omega_{\varepsilon, \delta}(m^{1/p - 1/2} \|A\|_1 / \sqrt{n})$. Here, $\|A\|_1$ is the Schatten-1 norm of $A$ and $\Omega_{\varepsilon, \delta}(\cdot)$ hides the dependency on the privacy parameters.*

To demonstrate the power of the above results, we obtain tight bounds for privately answering *prefix sum* and *parity queries*. These are two important classes of queries: for example, prefix sum is used as a subroutine in private learning (Kairouz et al., 2021) and parity queries[2] are often used for hardness results (Kasiviswanathan et al., 2011).

1. (Prefix sum) In this problem, the data curator outputs $\sum_{i \leq t} x_i$ of a vector $x = (x_1, x_2, \cdots, x_n)$ in a differentially private manner for all $t \leq n$. This is equivalent to asking linear queries with $A_{\text{prefix}} \in \{0, 1\}^{n \times n}$ where $A_{\text{prefix}}$ is a lower-triangular matrix with non-zero entry equal to one. In Theorem 1.5, we show that, for all constant $p$, the $\ell_p^p$-error of prefix sum under $(\varepsilon, \delta)$-differential privacy is $\Theta_{\varepsilon, \delta}(n^{1/p} \log(n))$ and can be achieved by the same mechanism for all $p = O(1)$; for $p = \omega(1)$, the gap between the upper and lower bound is of factor $\sqrt{\log(n)}$. This generalizes the result of Dwork et al. (2010) and Henzinger et al. (2023).

2. (Parity Queries). Let $\mathcal{Q}_{d,w}^P = \{q_P(x) = \prod_{i \in P} x_i : P \subset [d], |P| = w\}$ be the class of parity queries over the input domain $\{-1, 1\}^d$. In Theorem 1.6, we show that for any $(\varepsilon, \delta)$-differentially private mechanism $\mathcal{M}$ that takes as input $d$ and $w$,

$$\text{err}_{\ell_p^p} \left( \mathcal{M}, \mathcal{Q}_{d,w}^P \right) = \Omega_{\varepsilon, \delta} \left( m^{1/2 + 1/p} \right)$$

---

[1] Interpolation plays a key role in functional analysis (Riesz, 1927; Thorin, 1948), and it is one of the primary reasons for the initial support for Riesz (1910; 1913)'s study of $\ell_p$-norm despite Minkowski's skepticism (Minkowski, 1913).

[2] Depending on which parity queries are made, the workload matrix would consist of a subset of rows of a normalized $n \times n$ Hadamard matrix

for $m = \binom{d}{w}$. Since $O_{\varepsilon,\delta}\left(m^{1/2+1/p} \min\{p, \log(m)\}\right)$ is the $\ell_p^p$ error of the trivial Gaussian mechanism, this is optimal whenever $\min\{p, \log(m)\} = O(1)$.

Our study is motivated and inspired by recent elegant work by Nikolov & Tang (2024), who proved the *instance optimality* of the matrix mechanism instantiated using *correlated Gaussian noise* for *unbiased mean estimation*. They considered the following error metric[3]:

$$\mathsf{err}_{NT}(\mathcal{M}, A) := \max_{x \in \mathbb{R}^n} \left(\mathbb{E}\left[\|\mathcal{M}(x) - Ax\|_p^2\right]\right)^{1/2}. \tag{2}$$

Nikolov & Tang (2024) showed that, with the privacy notion where $\|x - x'\|_1 \leq O(1)$ *and* that $\sum_{i \in [n]}(x_i - x_i') = 0$ (commonly known as the *substitution model* of privacy), matrix mechanism is instance-optimal for any *unbiased* mechanism under the metric defined in eq. (2)[4]. Our work instead focuses on obliviousness of the matrix mechanism, a more natural $\ell_p^p$ metric and a different (stronger) privacy notion, i.e., it differs both in the error metric and the results:

1. To understand the difference between these two error metrics, consider the error vector $v \in \mathbb{R}^m$. The metric used in Nikolov & Tang (2024) amounts to estimating $\mathbb{E}[(|v_1|^p + \cdots + |v_m|^p)^{2/p}]$ instead of a more natural $\mathbb{E}[v_1^p + \cdots + v_m^p]$ in eq. (1). In other words, it does not explain the behavior of the error even in the case of the naive additive noise mechanisms. This is one of the primary reasons we believe our error metric is more natural.

2. They focused on instance optimality of *unbiased mean estimation*. While this is a well-studied problem, it does not cover the question of the $\ell_p$-optimality of general linear queries under the error metric defined by eq. (1) for general mechanisms. We answer this question broadly and prove equivalent results for a more natural error metric.

3. Differential privacy in the add/remove model (i.e., $\|x - x'\|_1 \leq 1$) is usually the most natural notion considered in literature of answering linear queries ( Edmonds et al. (2020); Nikolov et al. (2013); Bhaskara et al. (2012)). In contrast, Nikolov & Tang (2024) considered the substitution model for unbiased mean estimation over some convex polytopes. The sensitivity polytope related to the substitution model can be substantially smaller than that of the add/remove model (see also Appendix B.3 for a more detailed discussion). As a result, deriving a new lower bound for the stronger add/remove model is needed.

From a pure analysis perspective (and as is often the case in mathematics) as well, one prefers a metric respecting the symmetry as shown in our choice of metric, the $\ell_p$-norm, and $F_p$ moments studied in the streaming literature. While both of the error metrics (eq. (1) and eq. (2)) converge to the same metric as $p \to \infty$ and when $p \to 2$, the mathematical object the sequence captures as a function of $p$ is vastly different. That is, our results complement that of Nikolov & Tang (2024).

## 1.1 OUR CONTRIBUTIONS

Our main result is a lower bound on general $(\varepsilon, \delta)$-differentially private mechanisms for answering linear queries in high privacy regimes in terms of certain *factorization norms* Nikolov & Tang (2024) defined below[5] :

$$\gamma_{(p)}(A) := \min_{LR=A} \left\{ \sqrt{\mathsf{tr}_{p/2}(LL^\top)} \|R\|_{1\to 2} \right\}, \quad \text{where} \quad \mathsf{tr}_p(U) := \begin{cases} \left(\sum_{i=1}^d U_{ii}^p\right)^{1/p} & p < \infty \\ \max_{i \in [d]} |U_{ii}| & p = \infty \end{cases}$$

is the *p-trace*. Equipped with this definition, we state our lower bound:

---

[3]Nikolov and Tang confirmed with us that they did not consider the metric considered in this paper.

[4]We note that a Gaussian distribution is entirely characterized by its first two moments and, at a high level, eq. (2) captures the variance of the $\ell_p$ norm of the zero mean vector representing the additive error.

[5]Let $\|B\|_{p \to q} = \min_{\|x\|_p=1} \|Bx\|_q$. Then two commonly studied factorization norms in privacy and functional analysis denoted by $\gamma_2(A)$ and $\gamma_F(A)$ are defined as $\gamma_2(A) = \min_{LR=A}\{\|L\|_{2\to\infty}\|R\|_{1\to2}\}$ and $\gamma_F(A) = \min_{LR=A}\{\|L\|_F\|R\|_{1\to2}\}$. Both these norms are special cases of $\gamma_{(p)}(\cdot)$ because when $p = 2$, $\mathsf{tr}_{p/2}(LL^\top) = \|L\|_F^2$ and when $p \to \infty$, then $\mathsf{tr}_\infty(LL^\top) = \max_{i \in [d]}(LL^\top)_{ii} = \|L\|_{2\to\infty}^2$.

**Theorem 1.4 (Lower bound for $(\varepsilon, \delta)$-DP)** *Fix any $n, m \in \mathbb{N}$, $\varepsilon \in (0, \frac{1}{2})$, $0 \leq \delta \leq 1$ and $p \in [2, \infty)$. For any query matrix $A \in \mathbb{R}^{m \times n}$, if a mechanism $\mathcal{M} : \mathbb{R}^n \to \mathbb{R}^m$ preserves $(\varepsilon, \delta)$-differential privacy, then there exists a universal constant $C'$,*

$$\mathrm{err}_{\ell_p^p}(\mathcal{M}, A) \geq \frac{(1 - \tilde{\delta})\gamma_{(p)}(A)}{C'\varepsilon}, \quad \text{where} \quad \tilde{\delta} = \frac{2e^{2\varepsilon}(e^{1/2} - 1)}{e^\varepsilon - 1}\delta.$$

Theorem 1.4 generalizes the result of Edmonds et al. (2020) for $p = 2$ to all $p \geq 2$. Our result can also be contrasted with the lower bound which uses discrepancy methods. It is known that the $\ell_\infty$-error of an $(\varepsilon, \delta)$-differentially private algorithm for linear queries is lower bounded by the *hereditary discrepancy* of the corresponding matrix $A \in \mathbb{R}^{m \times n}$ (Muthukrishnan & Nikolov, 2012), which in turn is lower bounded by $\gamma_{(\infty)}/\sqrt{\log(m)}$ using its characterization in terms of a semidefinite program (Matoušek et al., 2020). Our result shows that we can get a $\sqrt{\log(m)}$ better lower bound. We complement this lower-bound with a tight upper bound in Appendix D (see Theorem D.1) matching it up to an $O(\log(1/\delta) \cdot \min\{p, \log(2m)\})$ factor, combining this and Theorem 1.4 gives Theorem 1.2.

The meaning of $\gamma_{(p)}(A)$ in Theorem 1.4 is not immediately apparent. Thus, as one of its applications, we study explicit lower bound (with respect to $n$ instead of $\gamma_{(p)}(A)$) for some special types of queries that are widely used in the community of privacy. We first characterize the accuracy of prefix sum, i.e., when the query matrix $A_{\mathsf{prefix}}$ is a lower-triangular all-one matrix. The upper bound in Theorem 1.5 follows from the *binary tree mechanism* (Chan et al., 2011; Dwork et al., 2010) while the lower bound uses Theorem 1.4. Notably, we can extend the lower bound for prefix sum queries to all $\varepsilon > 0$, rather than limiting it to a high privacy regime of $\varepsilon < 1/2$ as in Theorem 1.4.

**Theorem 1.5** *For any $n \in \mathbb{N}$ and any $p \in [2, \infty)$, the matrix mechanism, $\mathcal{M}_{\mathsf{fact}}$, achieves the following error guarantee while preserving $(\varepsilon, \delta)$-differential privacy:*

$$\mathrm{err}_{\ell_p^p}(\mathcal{M}_{\mathsf{fact}}, A_{\mathsf{prefix}}, n) = O\left(\frac{n^{1/p}\log(n)\sqrt{\log(1/\delta) \cdot \min\{p, \log(n)\}}}{\varepsilon}\right)$$

*Further, there is no $(\varepsilon, \delta)$-differentially private mechanism $\mathcal{M}$ that achieves*

$$\mathrm{err}_{\ell_p^p}(\mathcal{M}, A_{\mathsf{prefix}}, n) = o\left(\frac{(1 - \delta)n^{1/p}\log(n)}{e^{3\varepsilon} - 1}\right) \quad \text{for} \quad \delta \leq \min\left\{\frac{1}{16}, \varepsilon^2, \Theta\left(\frac{\varepsilon n^{\frac{2-p}{2p}}}{\ln(n)}\right)\right\}.$$

We note that $e^{3\varepsilon} - 1 = O(\varepsilon)$ in a high privacy regime where $\varepsilon = O(1)$, so the lower and upper bound match in such a regime. Theorem 1.5 recovers the result in Henzinger et al. (2023) for $p = 2$ as a special case. Moreover, it *exactly* characterizes the error of prefix sum with respect to any $\ell_p^p$ metric for all constant $p$. One can obtain an $\Omega(\log(n))$ lower bound on $\ell_\infty$-error using a slight modification of the packing argument in Dwork et al. (2010) for $\delta = o(1/n)$. Our bound extends the packing-based lower bound to larger values of $\delta$ and all $p \in [2, \infty)$.

As another application, in Theorem 1.6 (shown in Appendix C), we characterize the lower bound on privately answering parity queries. The theorem recovers the lower bound in Section 8 of Henzinger et al. (2023) for $p = 2$ and Section 3.6 of Edmonds et al. (2020) when $p \to \infty$.

**Theorem 1.6** *Let $\mathcal{Q}_{d,w}^P$ be the collection of parity queries. For any $(\varepsilon, \delta)$-differentially private mechanism $\mathcal{M}$ for answering queries in $\mathcal{Q}_{d,w}^P$, the worst case $\ell_p^p$ error*

$$\mathrm{err}_{\ell_p^p}\left(\mathcal{M}, \mathcal{Q}_{d,w}^P, \binom{d}{w}\right) = \Omega\left(\frac{(1 - \delta)}{e^{3\varepsilon} - 1}\binom{d}{w}^{1/2 + 1/p}\right).$$

**Organization of the proof.** In Section 2.1, we first develop the lower bound for additive noise mechanism on arbitrary matrix with linearly independent rows. For general matrix, in Section 2.2, we remove the linear independency assumption in the start of Section 2, and then with the help of the back-box reduction from additive noise mechanism to general mechanism, we give a $(\varepsilon, \delta)$-DP lower bound with respect to *general* matrix $A \in \mathbb{R}^{m \times n}$ in only high privacy regime, which establishes Theorem 1.4. Next, in Section 2.3, we prove our easy-to-use bound Theorem 1.3 based on Theorem 1.4. We derive Theorem 1.5 as a corollary of previous sections. The proof of Theorem 1.6 follows a similar reasoning and we defer it in Section C.

## 2 LOWER BOUND FOR $(\varepsilon, \delta)$-DP AND ITS APPLICATION

In this section, we prove our lower bound and its applications in proving lower bounds of prefix sum and parity queries in $\ell_p^p$ metric. Throughout this paper, we write $a \gtrsim b$ if there exists some universal constant $c$ such that $a \geq \frac{1}{c} b$. For proving the lower bound in terms of $(\varepsilon, \delta)$-differential privacy, we first consider a special class of mechanisms that adds noise sampled from an appropriate distribution to the real answer of the queries (a high-level idea of our proof is presented in Appendix B). We call such a class of mechanisms the *additive noise mechanisms*. Unlike Nikolov & Tang (2024), we do not assume that the mechanism is unbiased which makes our analysis more subtle.

Before stating the result, we fix some notations. Let $B_p^n := \{x \in \mathbb{R}^d : \|x\|_p \leq 1\}$ denote the $n$-dimensional $\ell_p$-ball and $AB_1^n := \{Ax : x \in B_1^n\}$ denote the *sensitivity polytope*. To describe the lower bound, for any matrix $A \in \mathbb{R}^{m \times n}$, we define the map, $\kappa : \mathbb{R}^{m \times n} \to \mathbb{R}$, that computes the width of the *sensitivity polytope* with respect to the most "narrow" direction:

$$\kappa(A) := \min_{\|\theta\|_2 = 1} w_{AB_1^n}(\theta) \quad \text{where} \quad w_{AB_1^n}(\theta) := \max_{\|x\|_1 \leq 1} \theta^\top Ax - \min_{\|x\|_1 \leq 1} \theta^\top Ax. \tag{3}$$

To prove Theorem 1.4, we first show a lower bound for additive noise mechanisms when $A$ has linearly independent rows. We then remove this assumption in a high privacy regime ($\varepsilon < 1/2$) in Section 2.2; Theorem 1.4 then follows by using a general reduction of Bhaskara et al. (2012).

### 2.1 LOWER BOUND ON ADDITIVE NOISE MECHANISMS

**Theorem 2.1 (Lower bound for additive noise mechanisms)** *Fix any* $\varepsilon > 0$, $p \in [2, \infty)$ *and query matrix* $A \in \mathbb{R}^{m \times n}$. *There exists a* $\delta(A, \varepsilon, n) := \min\left\{ \frac{1}{16}, \varepsilon^2, \frac{\varepsilon \cdot \kappa(A) n^{1-2/p}}{12 \gamma_{(p)}(A)} \right\}$ *such that for any* $\delta \leq \delta(A, \varepsilon, n)$, *if* $\mathcal{M}$ *is a* $(\varepsilon, \delta)$-*differentially private additive noise mechanism, then for any* $x \in \mathbb{R}^n$,

$$\left( \mathbb{E}\left[ \|\mathcal{M}(x) - Ax\|_p^p \right] \right)^{1/p} \geq \frac{(1 - \delta')\gamma_{(p)}(A)}{8(e^{3\varepsilon} - 1)}, \quad \text{where} \quad \delta' = \frac{2\delta}{1 - e^{-\varepsilon}}.$$

The above theorem implies an almost tight lower bound in a high privacy regime. For example, when $0 \leq \varepsilon \leq \frac{1}{3}$, since $3\varepsilon \leq e^{3\varepsilon} - 1 \leq 6\varepsilon$, it directly implies that

$$\left( \mathbb{E}\left[ \|\mathcal{M}(x) - Ax\|_p^p \right] \right)^{1/p} \geq \frac{(1 - \delta')\gamma_{(p)}(A)}{48\varepsilon}.$$

This matches the upper bound given in Theorem D.1. For additive noise mechanisms, Theorem 2.1 is naturally instance-optimal on any $x \in \mathbb{R}^n$. We note that the range of $\delta$ in Theorem 2.1 depends on $\kappa(A)$, and it is easy to verify that $\kappa(A) > 0$ if and only if $A$ has linearly independent rows (see Lemma 2.5 for details). While special linear queries such as prefix sum and parity queries inherently possess linearly independent rows, there are many interesting matrices without linearly independent rows. In the high privacy regime, which was the setting considered in Edmonds et al. (2020), we remove the full rank assumption (see Theorem 2.8 in Section 2.2). This underpins Theorem 1.4.

The main technical obstacle of Theorem 2.1 lies in making explicit all the intricate dependencies on the width of the sensitivity polytope, and how to handle the bias in lower bound proofs. We note that Edmonds et al. (2020) only studies $\ell_2^2$ error. Therefore, without loss of generality, it can be assumed that the bias is 0. Nikolov & Tang (2024) studies an unbiased setting and their approximate DP lower bound depends on the minimum width of the polytope ($w_0$ in Nikolov & Tang (2024)). Our lower bound does not assume unbiasedness, and our lower bound in Theorem 2.1 does not depend on the minimum width in the bound itself. Instead, the minimum width is only required in Theorem 2.1 for the applicable range of $\delta$. This means that our bound remains non-trivial even if the minimum width is like $1/n$. In proving the new lower bound, we also adapt geometric characterizations in Nikolov & Tang (2024) to handle bias of an additive noise mechanism. We also give a more detailed discussion in Appendix B.3.

To prove Theorem 2.1, we consider mechanisms of the form $\mathcal{M}(x) = Ax + z$, where $z$ is stochastically independent of $x$. For any input $x \in \mathbb{R}^n$, we define the covariance matrix of $\mathcal{M}(x)$ to be

$$\Sigma_{\mathcal{M}}(x) = \mathbb{E}[(\mathcal{M}(x) - \mathbb{E}[\mathcal{M}(x)])(\mathcal{M}(x) - \mathbb{E}[\mathcal{M}(x)])^\top].$$

Since an additive noise mechanism can be biased, $\mathbb{E}[\mathcal{M}(x)]$ is not necessarily $Ax$. We prove in Appendix E.1 the following relationship between the $\ell_p^p$ error and the $p$-trace of the covariance matrix.

**Lemma 2.2** *Fix any $p \in [2, \infty)$ and any additive noise mechanism $\mathcal{M} : \mathbb{R}^n \to \mathbb{R}^m$. It holds that*

$$\forall \in \mathbb{R}^n, \quad \left( \mathbb{E}\left[ \|\mathcal{M}(x) - Ax\|_p^p \right] \right)^{1/p} \geq \sqrt{\mathsf{tr}_{p/2}(\Sigma_{\mathcal{M}}(x))}.$$

Therefore to prove Theorem 2.1, it suffices to prove a lower bound on $\mathsf{tr}_{p/2}(\Sigma_{\mathcal{M}}(x))$ for any additive noise private mechanism $\mathcal{M}(\cdot)$. To start with, we give a statement bounding the bias of an additive noise mechanism. In particular, using the Hölder's inequality, for $p \geq 2$, we have

$$\|\mathbb{E}z\|_2^2 = \sum_{i \in [n]} (\mathbb{E}z_i)^2 \leq \left( \sum_{i \in n} (\mathbb{E}z_i)^p \right)^{\frac{2}{p}} \cdot n^{(p-2)/p} \leq \left( \mathbb{E}[\|z\|_p^p] \right)^{2/p} \cdot n^{(p-2)/p}.$$

Taking the square root of both sides gives the following result.

**Lemma 2.3** *Fix $p \geq 2$. Let $\mathcal{M}(x) = Ax + z$ be an additive noise mechanism with $z \in \mathbb{R}^m$, then*

$$\left( \mathbb{E}[\|\mathcal{M}(x) - Ax\|_p^p] \right)^{1/p} \geq \|\mathbb{E}z\|_2 \cdot n^{(1/p-1/2)}.$$

Therefore, we can assume $\|\mathbb{E}[z]\|_2 \leq \frac{\gamma_{(p)}(A) n^{(p-2)/2p}}{\varepsilon}$. Otherwise, due to Lemma 2.3, for all $x \in \mathbb{R}^n$,

$$\left( \mathbb{E}\left[ \|\mathcal{M}(x) - Ax\|_p^p \right] \right)^{1/p} \geq \frac{\|\mathbb{E}[z]\|_2}{n^{(p-2)/2p}} > \frac{\gamma_{(p)}(A) n^{(p-2)/2p}}{\varepsilon n^{(p-2)/2p}} = \frac{\gamma_{(p)}(A)}{\varepsilon} > \frac{(1-\delta')\gamma_{(p)}(A)}{\varepsilon}.$$

So, it suffices to prove a lower bound on $\mathsf{tr}_{p/2}(\Sigma_{\mathcal{M}}(x))$ for additive noise mechanisms with small bias. For this, we use a folklore trick (Smith, 2016) that has been used frequently in the literature of differential privacy. It consists of the following steps: For distributions $\mathcal{D}$ and $\bar{\mathcal{D}}$ corresponding to the output distribution of a privacy-preserving mechanism on the neighboring dataset, we first define the support on which the privacy loss variable with respect to $\mathcal{D}$ and $\bar{\mathcal{D}}$ is bounded. Then we update the probability distribution $\mathcal{D}$ such that the privacy loss random variable with respect to the new distribution and $\bar{\mathcal{D}}$ is still bounded and the measure of the new distribution is close in some metric to $\mathcal{D}$. In more details, for any two distributions $P$ and $Q$ over $\Omega$ and $\varepsilon > 0$, let $S_{P,Q,\varepsilon} := \left\{ \omega \in \Omega : e^{-\varepsilon} \leq \frac{P(\omega)}{Q(\omega)} \leq e^{\varepsilon} \right\}$ be the subset of the ground set $\Omega$ in which $P$ and $Q$ are $\varepsilon$-indistinguishable. Note that this is the same as $Bad_0$ in Kasiviswanathan & Smith (2014b). As in Nikolov & Tang (2024), define

$$\widehat{P} = \frac{Q(S_{P,Q,2\varepsilon})}{P(S_{P,Q,2\varepsilon})} P(T \cap S_{P,Q,2\varepsilon}) + Q(T \backslash S_{P,Q,2\varepsilon}) \tag{4}$$

where $T \subseteq \Omega$. This allows us to reduce differential privacy to $\chi^2$-*divergence* (eq. (10)) using Lemma 46 in Nikolov & Tang (2024) (see Lemma A.18). The following lemma (proven in Appendix E.2) states that, for a small bias additive noise mechanism, if $\Omega \subseteq \mathbb{R}$ and $P,Q$ are distributions of some additive noise mechanism on neighboring datasets, then $|\mathbb{E}_{X \sim \widehat{P}}[X] - \mathbb{E}_{X \sim Q}[X]|$ cannot be small.

**Lemma 2.4** *Suppose the additive noise mechanism $\mathcal{M}(x) = Ax + z$ is $(\varepsilon, \delta)$-differentially private. Fix any $\theta \in \mathbb{R}^m$. Let $M_\theta(x) : \mathbb{R}^n \to \mathbb{R}$ such that $M_\theta(x) := \theta^\top Ax + \theta^\top z$ where $\|\mathbb{E}[z]\|_2 \leq \frac{\gamma_{(p)}(A)}{\varepsilon} n^{(p-2)/2p}$. Let $\varepsilon, \delta$ be such that $\delta' = \frac{2\delta}{1-e^{-\varepsilon}} \leq \min\{\frac{1}{16}, \frac{\varepsilon \cdot \kappa(A) \cdot n^{\frac{p-2}{2p}}}{12\gamma_{(p)}(A)}, 1-e^{-\varepsilon}\}$. Then, for any $x \in \mathbb{R}^n$, there exists a neighboring dataset $x'$ such that if $P, Q$ are the distributions of $M_\theta(x)$ and $M_\theta(x')$ respectively, and let $\widehat{P}$ be the distribution defined in eq. (4), we have*

$$|\mathbb{E}_{X \sim \widehat{P}}[X] - \mathbb{E}_{X \sim Q}[X]| \geq \left( \frac{1}{2} - 2\delta' \right) \cdot \frac{w_{AB_1^n}(\theta)}{2} - \frac{17}{8}\sqrt{\delta' \mathsf{Var}[\theta^\top \mathcal{M}(x)]}.$$

We will use Lemma 2.4 to prove Theorem 2.1. To do so, we need to study the applicable range of $\delta'$ in Lemma 2.4. Fix any $\theta \in \mathbb{R}^m$. Given any $\varepsilon \in (0, \frac{1}{2})$, let $\delta(A, \varepsilon, n)$ be the maximum value of $\delta$ such that

$$\delta' = \frac{2\delta}{1-e^{-\varepsilon}} \leq \min \left\{ \frac{1}{16}, 1-e^{-\varepsilon}, \frac{\varepsilon \cdot \kappa(A) \cdot n^{\frac{2-p}{2p}}}{12\gamma_{(p)}(A)} \right\}.$$

Note that $\delta' > 0$ iff $\kappa(A) > 0$ as other quantities are positive. We characterize when $\kappa(A) > 0$ in Appendix F.2 that ensures $\delta' > 0$ through the following lemma:

**Lemma 2.5** $\kappa(A) > 0$ *if and only if $A$ has linearly independent rows.*

We will also need two geometric lemmas inspired by Nikolov & Tang (2024), that connect $\ell_1$ geometry and $\ell_2$ geometry, and also to the factorization norm. For $K, L \subseteq \mathbb{R}^m$, denote by $K \subseteq_\leftrightarrow L \Leftrightarrow \exists v \in \mathbb{R}^m, K + v \subseteq L$. That is, $K \subseteq_\leftrightarrow L$ means that $K$ is covered by $L$ in terms of translation. We define

$$\Lambda_p(A) := \inf_{W \in \mathbb{R}^{m \times m}} \left\{ \sqrt{\mathsf{tr}_{p/2}(WW^\top)} : AB_1^n \subseteq_\leftrightarrow WB_2^m \right\}.$$

The first lemma is similar to the one in Nikolov & Tang (2024), but for a general mechanism (instead of only for unbiased mechanisms). This lemma shows that if the variance of one way marginal of an additive noise mechanism $\mathcal{M}(\cdot)$ is lower bounded by the square of the width of the sensitivity polytope $AB_1^n$, then $AB_1^n$ can be covered by $C\sqrt{\Sigma_\mathcal{M}(x)}B_2^m$ in terms of translation with proper $C$.

**Lemma 2.6 (Nikolov & Tang (2024))** *Let $\mathcal{M} : \mathbb{R}^n \to \mathbb{R}^m$ be any randomized mechanism and $A \in \mathbb{R}^{m \times n}$ be any matrix. If there exists some universal constant $C$ such that for any input $x \in \mathbb{R}^n$ and any $\theta \in \mathbb{R}^m$, it satisfies $\mathsf{Var}[\theta^\top \mathcal{M}(x)] \geq \left( \frac{w_\theta(AB_1^n)}{C} \right)^2$, then $AB_1^n \subseteq_\leftrightarrow C\sqrt{\Sigma_\mathcal{M}(x)}B_2^m$.*

The original lemma in Nikolov & Tang (2024) is only stated for unbiased mechanisms instead of general mechanisms. Thus, we include a proof in Appendix F (restated as Lemma F.4) for completeness.

The final piece we need is a lemma implicit in Nikolov & Tang (2024) that connects $\Lambda_p(A)$ and the factorization norm $\gamma_{(p)}(A)$.

**Lemma 2.7 (Nikolov & Tang (2024))** *For any $p \in [2, \infty]$ and $A \in \mathbb{R}^{m \times n}$, $\Lambda_p(A) \geq \gamma_{(p)}(A)$.*

Now we are ready to prove Theorem 2.1.

**Proof:** [Proof of Theorem 2.1]. Let $\tilde{\varepsilon} = 2\varepsilon - \log(1 - \delta')$. Note that, for every $\varepsilon > 0$, we have $\delta' \leq 1 - e^{-\varepsilon}$, and thus $\tilde{\varepsilon} \leq 2\varepsilon + \varepsilon \leq 3\varepsilon$. Finally $n^{1-2/p} \geq n^{-1}$. For any $x$ and $x'$ chosen in Lemma 2.4, we consider two cases based on the variance, $\mathsf{Var}[\theta^\top \mathcal{M}(x)]$:

1. **When** $\mathsf{Var}[\theta^\top \mathcal{M}(x)] < \frac{\left( w_{AB_1^n}(\theta) \right)^2}{256\delta'}$. By Lemma 2.4, $|\mathbb{E}_{X \sim \widehat{P}}[X] - \mathbb{E}_{X \sim Q}[X]|$ is at least

$$\left( \frac{1}{2} - 2\delta' \right) \frac{w_{AB_1^n}(\theta)}{2} - \frac{17}{8}\sqrt{\delta' \mathsf{Var}[\theta^\top \mathcal{M}(x)]} \geq \frac{1 - 8\delta'}{8} w_{AB_1^n}(\theta).$$

   Note that $Q$ is the distribution of $\theta^\top \mathcal{M}(x')$, and $\mathsf{Var}[\theta^\top \mathcal{M}(x)] = \mathsf{Var}[\theta^\top \mathcal{M}(x')]$ since the oblivious noise $\theta^\top z$ is independent of the input. Then, by the Hammersley-Chapman-Robins bound (Lemma A.14), we have that for such a pair of datasets $(x, x')$:

$$\mathsf{Var}[\theta^\top \mathcal{M}(x)] = \mathsf{Var}[\theta^\top \mathcal{M}(x')] \geq \frac{\left| \mathbb{E}_{X \sim \widehat{P}}[X] - \mathbb{E}_{X \sim Q}[X] \right|^2}{\chi^2(\widehat{P}, \theta^\top \mathcal{M}(x'))} \geq \frac{(1 - 8\delta')^2 \left( w_{AB_1^n}(\theta) \right)^2}{64\chi^2(\widehat{P}, Q)}$$

$$\geq \frac{(1 - 8\delta')^2 \left( w_{AB_1^n}(\theta) \right)^2}{64e^{-\tilde{\varepsilon}}(e^{\tilde{\varepsilon}} - 1)^2}. \tag{5}$$

   Here, we used Lemma A.18 that shows that $\widehat{P}$ and $Q$ are $\tilde{\varepsilon}$-*indistinguishable* (Definition A.17). Thus $\chi^2(\widehat{P}, Q) \leq e^{-\tilde{\varepsilon}}(e^{\tilde{\varepsilon}} - 1)^2$ by Lemma 39 in Nikolov & Tang (2024) (also see Lemma A.13).

2. **When** $\mathsf{Var}[\theta^\top \mathcal{M}(x)] \geq \frac{\left( w_{AB_1^n}(\theta) \right)^2}{256\delta'}$. First note that, when $\delta' \leq \varepsilon^2 \leq \tilde{\varepsilon}^2$, we have $\frac{1 - 8\delta'}{16\tilde{\varepsilon}} \leq \frac{1}{16\sqrt{\delta'}}$. Therefore, for every $\theta \in \mathbb{R}^m$, as in the other case, for any $x$,

$$\mathsf{Var}[\theta^\top \mathcal{M}(x)] \geq \frac{(1 - 8\delta')^2 \left( w_{AB_1^n}(\theta) \right)^2}{64e^{-3\varepsilon}(e^{3\varepsilon} - 1)^2}. \tag{6}$$

Lemma 2.6 with eq. (5) and eq. (6) implies that $AB_1^n \subseteq_\leftrightarrow C\sqrt{\Sigma_\mathcal{M}(x)}B_2^m$ where $C = \frac{16\tilde\varepsilon}{1-8\delta'}$. So

$$C^2 \cdot \mathsf{tr}_{p/2}(\Sigma_\mathcal{M}(x)) \geq \inf_{W \in \mathbb{R}^{m \times m}} \left\{ \mathsf{tr}_{p/2}(WW^\top) : AB_1^n \subseteq_\leftrightarrow WB_2^m \right\} = (\Lambda_p(A))^2 \qquad (7)$$

for all $p \geq 2$. That is, $\mathsf{tr}_{p/2}(\Sigma_\mathcal{M}(x)) \geq \left( \frac{(1-8\delta')\Lambda_p(A)}{16\tilde\varepsilon} \right)^2$.

Combining Lemma 2.2, Lemma 2.7, and eq. (7) therefore gives us the result:

$$\left( \mathbb{E} \left[ \|\mathcal{M}(x) - Ax\|_p^p \right] \right)^{1/p} \geq \sqrt{\mathsf{tr}_{p/2}(\Sigma_M(X))} \geq \frac{(1-8\delta')\gamma_{(p)}(A)}{16\tilde\varepsilon}.$$

The proof of Theorem 2.1 is complete after replacing $\tilde\varepsilon$ by $\varepsilon$. $\qquad\square$

## 2.2 PROOF OF THEOREM 1.4

Theorem 2.1 assumes that rows of the linear query matrix is linearly independent (i.e., $\kappa(A) > 0$), otherwise the lower bound reduces to the one for pure differential privacy. We next show that for additive noise mechanisms, this assumption can be removed in the most natural high privacy regime:

**Theorem 2.8** *Fix any* $0 < \varepsilon < \frac{1}{2}$, $0 \leq \delta \leq 1$, $p \in [2, \infty]$ *and query matrix* $A \in \mathbb{R}^{m \times n}$. *If* $M(\cdot)$ *is an additive noise mechanism such that* $\mathcal{M}(x) = Ax + z$ *for any dataset* $x \in \mathbb{R}^n$ *and* $M(\cdot)$ *preserves* $(\varepsilon, \delta)$-*differential privacy, then for every* $x \in \mathbb{R}^n$, *we have that there exists a universal constant* $C$,

$$\left( \mathbb{E} \left[ \|\mathcal{M}(x) - Ax\|_p^p \right] \right)^{1/p} \geq \frac{(1-\delta')\gamma_{(p)}(A)}{C\varepsilon}, \quad \text{where} \quad \delta' = \frac{e^{1/2}-1}{1-e^{-\varepsilon}}\delta.$$

Unlike the proof of Theorem 2.1, we prove the above result using Lemma F.5, in which some of the technical ingredients are implicit in (Kasiviswanathan et al., 2010, Lemma 4.12). Then we combine it with (Nikolov & Tang, 2024, Lemma 35). We defer the proof of Theorem 2.8 to Section F. Note that for general $\varepsilon > 0$, the analysis of Theorem 2.1 also naturally gives an $\Omega(\gamma_{(p)}(A)/(e^{3\varepsilon} - 1))$ lower bound when $A$ has full rank rows, while Theorem 2.8 only works for high privacy regime.

To obtain a lower bound for general $(\varepsilon, \delta)$-differentially private algorithm, we recall that the reduction in Bhaskara et al. (2012) does not rely on the error metric. In particular, Theorem 1.4 follows by combining Theorem 2.8 and the reduction given by the following theorem to get a worst-case lower bound for arbitrary mechanisms.

**Theorem 2.9 (Theorem 4.3 in Bhaskara et al. (2012))** *Fix any* $A \in \mathbb{R}^{m \times n}$. *Let* $\mathcal{M} : \mathbb{R}^n \to \mathbb{R}^m$ *be a* $(\varepsilon, \delta)$-*differentially private algorithm. Then there exists a* $(2\varepsilon, e^\varepsilon \delta)$-*differentially private algorithm* $\mathcal{M}' := Ax + z$ *with oblivious* $z$ *such that* $\mathsf{err}_{\ell_p^p}(\mathcal{M}', A) \leq \mathsf{err}_{\ell_p^p}(\mathcal{M}, A)$.

## 2.3 CONNECTING $\gamma_{(p)}(\cdot)$ AND SCHATTEN-1 NORM: PROOF OF THEOREM 1.3

In previous sections, we established the connection between the hardness of privately answering linear queries defined by $A$ and the generalized factorization norm of $A$, denoted as $\gamma_{(p)}(A)$. However, expressing $\gamma_{(p)}(A)$ analytically for a general matrix $A$ can be difficult. To provide a more practical lower bound and facilitate potential applications, in the following lemma, we give a lower bound of $\gamma_{(p)}$ in terms of the Schatten-1 norm of $A$, which is simply the sum of singular values of $A$.

**Lemma 2.10** *Let* $A \in \mathbb{R}^{m \times n}$ *be any real matrix. It holds that*

$$\gamma_{(p)}(A) \geq m^{1/p}\|A\|_1/\sqrt{mn}.$$

**Proof:** By (Nikolov & Tang, 2023, Theorem 23) and Lemma 27 in Nikolov & Tang (2024), for any $p > 2$, the $\gamma_{(p)}$-norm of $A$ can be rewritten as the following optimization problem:

$$\gamma_{(p)}(A) = \max\{\gamma_{(2)}(DA) : D \text{ is diagonal}, D \succeq 0, \mathrm{Tr}_q(D^2) = 1\}$$

where $q = \frac{p}{p-2}$. Let $D = m^{\frac{1}{p}-\frac{1}{2}}I$, then $D$ is a diagonal PSD matrix and $\mathrm{Tr}_q(D^2) = m^{\frac{2}{p}-1} \cdot m^{\frac{1}{q}} = 1$. Using (Henzinger et al., 2023, Lemma 1.1), we therefore have

$$\gamma_{(p)}(A) \geq m^{1/p-1/2} \cdot \gamma_{(2)}(I \cdot A) = m^{1/p-1/2} \cdot \gamma_{(2)}(A) \geq \frac{m^{1/p-1/2}\|A\|_1}{\sqrt{n}},$$

completing the proof. □

Theorem 1.3 directly follows from Lemma 2.10 and Theorem 1.4.

## 2.4 APPLICATION I: TIGHT LOWER BOUND FOR PRIVATE PREFIX SUM WITH $\ell_p^p$ ERROR

So far, we have seen that the lower bounds on privately answering linear queries depend on $\gamma_{(p)}(A)$. In this section, we focus on a fundamental type of query: prefix sum and establish an explicit bound that underpins Theorem 1.5 by giving tight upper and lower bounds of $\gamma_{(p)}(A)$ and $\kappa(A)$ for such a specific $A$. In particular, given $n \in \mathbb{N}_+$, we consider the prefix sum (i.e., continual counting) matrix $A_{\mathsf{prefix}}$, whose $(i,j)$-th entry is

$$A_{\mathsf{prefix}}[i,j] = \begin{cases} 1 & i \geq j \\ 0 & \text{otherwise} \end{cases} \tag{8}$$

be the matrix computing prefix sum of the dataset $x \in \mathbb{R}^n$. We first give the following lower bound on private prefix sum:

**Theorem 2.11** *Fix any $\varepsilon \in (0, \frac{1}{6})$ and $p \in [2, \infty]$. Then, for any $\delta \leq C_\varepsilon n^{1/p-1/2}$ where*

$$C_\varepsilon = \min\left\{ \frac{1}{12} \frac{\varepsilon(1-e^{-\varepsilon})e^{-\varepsilon}}{(1+\ln(4n/5)/\pi)}, \frac{\varepsilon^2 e^{-\varepsilon}(1-e^{-\varepsilon})}{2} \right\},$$

*if $M : \mathbb{R}^n \to \mathbb{R}^m$ preserves $(\varepsilon, \delta)$-differential privacy, then*

$$\max_{x \in \mathbb{R}^n} \left( \mathbb{E}\left[ \|\mathcal{M}(x) - A_{\mathsf{prefix}}x\|_p^p \right] \right)^{1/p} \geq (1-\tilde{\delta}) \cdot \frac{n^{1/p}\log n}{96\varepsilon} \quad \text{where} \quad \tilde{\delta} = \frac{2\delta e^\varepsilon}{(1-e^{-\varepsilon})}.$$

Next, we show that there exists a factorization of $A_{\mathsf{prefix}}$ such that the $\ell_p^p$-error of the matrix mechanism is bounded by $O(n^{1/p}\log(n))$ implying the lower bound in Theorem 2.11 is optimal for $p = O(1)$ proving Theorem 1.5. If $p = \Omega(1)$, then this lower bound is near optimal with only a $\Theta(\sqrt{\log n})$ gap.

**Theorem 2.12** *Fix parameters $\varepsilon > 0$ and $0 < \delta < 1$. Given any $x \in \mathbb{R}^n$, there exists a $(\varepsilon, \delta)$-differentially private matrix mechanism $M$ such that*

$$\left( \mathbb{E}[\|M(x) - A_{\mathsf{prefix}}x\|_p^p] \right)^{1/p} \leq \frac{3n^{1/p}\log n}{\varepsilon} \sqrt{\frac{\log(1/\delta) \cdot \min\{p, \log(n)\}}{2}}.$$

The upper and lower bound in Theorem 1.5 directly follows from Theorem 2.11 and Theorem 2.12. We defer the proof of Theorem 2.12 to Appendix G.2.

### 2.4.1 PROOF OF THEOREM 2.11

To prove this theorem for all $\varepsilon > 0$, we need two things: firstly, a lower bound on the factorization norm $\gamma_{(p)}(A_{\mathsf{prefix}})$; secondly, in order to determine $\delta'$, we show that $\kappa(A_{\mathsf{prefix}})$ is lower bounded by a constant (Lemma 2.14). Such a geometric property of $A_{\mathsf{prefix}}$ could also be of independent interest. Then, the explicit lower bound on privately computing $A_{\mathsf{prefix}}x$ is obtained by applying Theorem 2.1 and the black-box reduction given in Theorem 2.9 (Theorem 4.3 in Bhaskara et al. (2012)).

**Lemma 2.13** *Let $A_{\mathsf{prefix}}$ be the matrix defined in Equation (8). Then, $\gamma_{(p)}(A_{\mathsf{prefix}}) \gtrsim n^{1/p}\log n$.*

**Proof:** Recall that for any $p > 2$, the $\ell_p$ factorization norm of $A_{\mathsf{prefix}}$ can be rewritten as:

$$\gamma_{(p)}(A_{\mathsf{prefix}}) = \max\{\gamma_{(2)}(DA_{\mathsf{prefix}}) : D \text{ is diagonal }, D \succeq 0, \mathrm{Tr}_q(D^2) = 1\}$$

where $q = \frac{p}{p-2}$. Let $D = n^{1/p-1/2}I$, then $D$ is a diagonal PSD matrix and $\mathrm{Tr}_q(D^2) = n^{2/p-1} \cdot n^{1/q} = 1$. Using (Henzinger et al., 2023, equation (5.30)),

$$\gamma_{(p)}(A_{\mathsf{prefix}}) \geq n^{1/p-1/2} \cdot \gamma_{(2)}(I \cdot A_{\mathsf{prefix}}) \gtrsim n^{1/p}\log n$$

completing the proof. □

Next, we compute $\kappa(A_{\mathsf{prefix}})$. The proof of Lemma 2.14 is given in Section G.1.

**Lemma 2.14** *Let $\kappa(\cdot)$ be as defined in eq.* (3). *Then $\kappa(A_{\mathsf{prefix}}) = 2$.*

Now we are ready to complete the proof of Theorem 2.11, which is a lower bound for private continual releasing of the prefix sum on arbitrary $\ell_p^p$ metric with $2 \leq p < \infty$.

Recalling Theorem 2.1, it remains to show $\delta'(A_{\mathsf{prefix}}, \varepsilon, n) \geq C_\varepsilon n^{1/p-1/2}$. In particular, we have the following:

$$\delta'(A_{\mathsf{prefix}}, \varepsilon, n) = \frac{2(e^\varepsilon - 1)}{e^{2\varepsilon}} \cdot \min\left\{\frac{1}{16}, \frac{\varepsilon \cdot \kappa(A_{\mathsf{prefix}}) \cdot n^{\frac{2-p}{2p}}}{12\gamma_{(p)}(A_{\mathsf{prefix}})}, \varepsilon^2\right\} \geq \frac{2(e^\varepsilon - 1)}{e^{2\varepsilon}} \min\left\{\frac{\varepsilon \cdot n^{\frac{2-p}{2p}}}{6\gamma_F(A_{\mathsf{prefix}})}, \varepsilon^2\right\}$$

$$\geq n^{\frac{2-p}{2p}} \cdot \min\left\{\frac{1}{12}\frac{\varepsilon(1 - e^{-\varepsilon})e^{-\varepsilon}}{(1 + \ln(4n/5)/\pi)}, \frac{\varepsilon^2 e^{-\varepsilon}(1 - e^{-\varepsilon})}{2}\right\} = \frac{C_\varepsilon}{n^{1/2-1/p}},$$

where the first inequality comes from that $\gamma_{(p)}(A) \leq \gamma_{(2)}(A) = \gamma_F(A)$ for any $A$ and $p \geq 2$, the second inequality follows from Henzinger et al. (2023). This completes the proof.

## 3 DISCUSSION

In this paper, we established lower bounds on approximating the linear query $Ax$ with respect to approximate differential privacy under $\ell_p^p$ error, so we can study the optimality of matrix mechanisms not only in expectation but also with respect to probability tail bounds. For limitations, we note that we only give a worst case lower bound over all $x \in \mathbb{R}^n$ by the definition of $\ell_p^p$ error metric (see also eq. (1)). To understand why we cannot get a instance-optimal lower bound, consider a trivial mechanism $M_{x_0}$ such that for any $x \in \mathbb{R}^n$, it always outputs $Ax_0$ where $x_0 \in \mathbb{R}^n$ is any given dataset. Clearly $M_{x_0}$ is not an oblivious additive noise mechanism, and it preserves perfect differential privacy, i.e., $\varepsilon = 0$, and perfect accuracy on the input $x_0$, which explains why an instance-optimal lower bound is unrealistic for general mechanisms.

In Nikolov & Tang (2024), the authors study unbiased mechanism, and show that the Gaussian mechanism is indeed instance-optimal over all such unbiased mechanisms, by giving an asymmetric lower bound saying that if an unbiased mechanism performs well in an input $x_0$, then it must perform worse in some other inputs $x'$ where $x'$ neighboring $x_0$. It is still open if such an asymmetric lower bound exists for general linear queries over all general mechanisms.

## ACKNOWLEDGMENTS

The authors would like to thank Aleksandar Nikolov for his valuable comments and especially his insights on removing the assumption of linear independence in Edmonds et al. (2020) and an earlier version of this paper. The authors also thank George Li for his discussions during the project and his helpful feedback on the paper draft.

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

# A BASIC DEFINITIONS AND PRELIMINARIES

**Matrix theory and Convex Geometry** We first introduce several definitions regarding the matrix norms and geometric properties of the query matrix $A$.

**Definition A.1 (Schatten-1 norm)** *Let $s_1, \cdots, s_m$ be the singular values of $A$, we define the Schatten-1 norm to be*

$$\|A\|_1 = \sum_{i=1}^m s_i.$$

**Definition A.2 ($p$-trace)** *Fix any $d \in \mathbb{N}_+$. Let $U \in \mathbb{R}^{d \times d}$ be a positive semi-definite matrix, we define the $p$-trace norm to be*

$$\mathsf{tr}_p(U) := \left( \sum_{i=1}^d U_{ii}^p \right)^{1/p}$$

Naturally, we define $\mathsf{tr}_\infty(U) = \max_{i \in [d]} |U_{ii}|$. The following definition for generalized factorization norm was firstly pointed out by Nikolov and Tang Nikolov & Tang (2024):

**Definition A.3** *For any $2 \le p \le \infty$ and $A \in \mathbb{R}^{m \times n}$, we define*

$$\gamma_{(p)}(A) := \min_{LR=A} \left\{ \sqrt{\mathsf{tr}_{p/2}(LL^\top)} \|R\|_{1 \to 2} \right\}.$$

It can be verified that $\gamma_{(2)}(A) = \gamma_F(A)$ and $\gamma_{(\infty)}(A) = \gamma_2(A)$. This is because when $p = 2$, $\mathsf{tr}_{p/2}(LL^\top) = \|L\|_F^2$ and when $p \to \infty$, then $\mathsf{tr}_\infty(LL^\top) = \max_{i \in [d]} (LL^\top)_{ii} = \|L\|_{2 \to \infty}^2$. We use the following result to connect the factorization norm and the Schatten-1 norm:

**Lemma A.4 (Henzinger et al. (2023) and Li & Miklau (2013))** *Let $A \in \mathbb{C}^{m \times n}$ be a complex matrix. Then*

$$\gamma_{(2)}(A) \ge \frac{\|A\|_1}{\sqrt{n}}$$

Using Theorem A.4, Henzinger et al. Henzinger et al. (2023) showed the following bound:

**Theorem A.5 (Henzinger et al. (2023))** *For any $n \in \mathbb{N}$, let $M_{\mathsf{count}}$ be a lower triangular matrix with all ones. Then*

$$\gamma_{(2)}(M_{\mathsf{count}}) \ge \frac{\sqrt{n}}{\pi} \left( 2 + \ln\left( \frac{2n+1}{5} \right) + \frac{\ln(2n+1)}{2n} \right)$$

**Definition A.6 (Parity Query)** *Let $d$ and $w$ be integer parameters and let the domain be $\mathcal{X} = \{\pm 1\}^d$. Then a parity query is a query that belongs to the family of queries*

$$\mathcal{Q}_{d,w} = \left\{ q_P(x) = \prod_{i \in P} x_i : P \subset \{1, \cdots, d\}, |P| = w \right\}. \tag{9}$$

**Definition A.7 (Hadamard matrix)** *Fix any integer $d \ge 1$, the $d$-th Hadamard matrix $H_d$ is a $2^d \times 2^d$ matrix*

$$\begin{bmatrix} H_{d-1} & H_{d-1} \\ H_{d-1} & -H_{d-1} \end{bmatrix}.$$

*When $d = 0$, $H_0 = [1]$.*

The following definitions are related to the geometry property of a query matrix.

**Definition A.8** *Fix any $d \in \mathbb{N}$. For any $K, L \subset \mathbb{R}^d$, we write*

$$K \subseteq_{\leftrightarrow} L \Leftrightarrow \exists v \in \mathbb{R}^d, K + v \subseteq L.$$

This is saying that if $K \subseteq_{\leftrightarrow} L$, then $K$ can be covered by $L$ by "relocating" the center of $K$. Next, we define the width of a convex body.

**Definition A.9 (Width of a convex body)** *Given any vector $\theta \in \mathbb{R}^m$, we define the width of any convex body $K \subseteq \mathbb{R}^m$ with respect to $\theta$ be*

$$w_K(\theta) := \max_{x \in K} \theta^\top x - \min_{x \in K} \theta^\top x.$$

We use $B_p^d$ to denote the unit ball of dimension $d$ with respect to the $\ell_p$ norm. Formally,

$$B_p^d = \{x \in \mathbb{R}^d : \|x\|_p \leq 1\}.$$

For matrix $W$ with $d$ columns, we also write $WB_p^d = \{Wx : x \in B_p^d\}$ to denote the *sensitivity polytope* of $W$ with respect to the $p$-th norm.

**Definition A.10 (Nikolov & Tang (2024))** *For any query matrix $A \in \mathbb{R}^{m \times n}$ and $p \in [2, \infty]$, we define*

$$\Lambda_p(A) := \inf_{W \in \mathbb{R}^{m \times m}} \left\{ \sqrt{\mathsf{tr}_{p/2}(WW^\top)} : AB_1^n \subseteq_{\leftrightarrow} WB_2^m \right\}.$$

Here, we give some insights about why $\Lambda_p(A)$ in Definition A.10 is useful for establishing the lower bound. Geometrically, $AB_1^n$ is exactly the convex body comprising differences between the ground truth output of any pair of neighboring datasets, $A(x - x')$ where $\|x - x'\| \leq 1$. Since $\Lambda_p(A)$ is the minimum trace norm of $WW^\top$ where $WB_2^m$ covers the sensitivity polytope $AB_1^n$, then, $\Lambda_p(A)$ can be interpreted as a specific kind of measurements on the volume of the body $WB_2^m$ that "covers" $A(x - x')$ over all pair of neighboring datasets.

Intuitively, if this volume gets larger, it is harder to preserve utility because the outputs of neighboring datasets will be far apart. Therefore, it gives a way to prove the lower bound by establishing a connection between the $\ell_p^p$ error and $\Lambda_p(A)$. The following lemma also reveals the relationship between $\Lambda_p(A)$ and the factorization norm $\gamma_{(p)}(A)$:

**Lemma A.11 (Nikolov & Tang (2024))** *For any $p \in [2, \infty]$ and $A \in \mathbb{R}^{m \times n}$, $\Lambda_p(A) \geq \gamma_{(p)}(A)$.*

Basically speaking, for any matrix $A \in \mathbb{R}^{m \times n}$, one can always find a factorization of $A = LR$ such that $\|R\|_{1 \to 2}$ is smaller than 1 and that $AB_1^n \subseteq_{\leftrightarrow} LB_2^m$. Then, taking the $L$ that minimizes $\sqrt{\mathsf{tr}_{p/2}(LL^\top)}$ yields the above lemma.

**Differential privacy.** Here, we first introduce Gaussian mechanism, which is the main component of the upper bound proof in this paper.

**Lemma A.12 (Gaussian mechanism)** *Fix any $0 \leq \varepsilon, \delta \leq 1$. Let $f : X \to Y$ be any deterministic function. If for all neighboring dataset $x, x'$, $\|f(x) - f(x')\|_2 \leq \Delta$, then $\mathcal{M}(x) = f(x) + z$ where $z \sim \mathcal{N}(0, \sigma^2 I)$ satisfies $(\varepsilon, \delta)$-differential privacy as long as $\sigma^2 \geq \frac{9\Delta^2 \log(1/\delta)}{2\varepsilon^2}$.*

As in Nikolov & Tang (2024), one of the necessary conditions of DP algorithms that we will consider is that DP algorithms preserve the $\chi^2$-divergence between neighboring datasets. For two distribution $P$ and $Q$, the $\chi^2$ divergence between them is

$$\chi^2(P, Q) := \mathbb{E}_{x \sim Q} \left[ \left( \frac{P(x)}{Q(x)} - 1 \right)^2 \right]. \tag{10}$$

It is not hard to verify (perhaps it is also well-known) the following lemma:

**Lemma A.13 (Lemma 39 in Nikolov & Tang (2024))** *Suppose $M$ is an $\varepsilon$-differentially private algorithm and $x$, $x'$ be two neighboring datasets such that $\|x - x'\|_1 \leq 1$. Let $P$ and $Q$ be the distributions of $\mathcal{M}(x)$ and $\mathcal{M}(x')$ respectively. Then*

$$\chi^2(P, Q) \leq e^{-\varepsilon}(e^\varepsilon - 1)^2.$$

The reason why we consider $\chi^2$ distribution is that the lower bound of the variance of a real random variable can be characterized by its $\chi^2$ divergence between another arbitrary random variable. This is the classical Hammersley-Chapman-Robins bound stated in the following lemma:

**Lemma A.14 (Hammersley-Chapman-Robins bound)** *For any two distributions $P, Q$ over real numbers and for $X, Y$ distributed, respectively, according to $P$ and $Q$, we have*

$$\sqrt{\mathsf{Var}(Y)} \geq \frac{|\mathbb{E}[X] - \mathbb{E}[Y]|}{\sqrt{\chi^2(P,Q)}}.$$

We also need the following lemma in this paper:

**Lemma A.15 (Lemma 4.4 in Kasiviswanathan et al. (2010))** *Let $w \in \mathbb{R}^n$ be any single query and $M'(x) := w^\top x + z'$ ($z' \in \mathbb{R}$) be any additive noise mechanism that is $(\varepsilon, 0)$-differentially private for any $0 < \varepsilon < 1$, then $\mathbb{E}[z^2] \gtrsim \frac{1}{\varepsilon^2}$ for some universal constant $C$.*

**Lemma A.16 (Kasiviswanathan & Smith (2014b))** *Let $M$ be any $(\varepsilon, \delta)$-differentially private mechanism, let $P$ be the distribution of $\mathcal{M}(x)$ and $Q$ be the distribution of $\mathcal{M}(x')$. Let*

$$S_{P,Q,\varepsilon} := \left\{ \omega \in \Omega : e^{-\varepsilon} \leq \frac{P(\omega)}{Q(\omega)} \leq e^{\varepsilon} \right\}. \tag{11}$$

*Then*

$$\max\left\{ \Pr[P \notin S], \Pr[Q \notin S] \right\} \leq \delta' = \frac{2\delta}{1 - e^{-\varepsilon}}.$$

Given two distribution $P$ and $Q$ and set defined by eq. (11), Nikolov & Tang (2024) defined a a distribution $\widehat{P}$ such that for any $T \subset \Omega$,

$$\widehat{P} = \frac{Q(S_{P,Q,2\varepsilon})}{P(S_{P,Q,2\varepsilon})} P(T \cap S_{P,Q,2\varepsilon}) + Q(T \backslash S_{P,Q,2\varepsilon}) \tag{12}$$

Here we define $(\varepsilon, \delta)$-indistinguishability:

**Definition A.17 (Kasiviswanathan & Smith (2014b))** *Let $\Omega$ be a ground set and $\mu_1, \mu_2$ be two distributions with support $\Omega_1 \subseteq \Omega$, $\Omega_2 \subseteq \Omega$ respectively. We say that $\mu_1$ and $\mu_2$ are $(\varepsilon, \delta)$-indistinguishable for $\varepsilon > 0$ and $\delta \in (0, 1)$ if for any $S \subseteq \Omega$, it holds that*

$$\mu_1(S) \leq \mu_2(S) \cdot e^{\varepsilon} + \delta \quad and \quad \mu_2(S) \leq \mu_1(S) \cdot e^{\varepsilon} + \delta.$$

*If $\delta = 0$, we also say $\mu_1$ and $\mu_2$ are $\varepsilon$-indistinguishable.*

We use the following lemma to characterize the relation between $\widehat{P}$ and $Q$:

**Lemma A.18 (Lemma 46 in Nikolov & Tang (2024))** *Let $P, Q$ be a pair of $(\varepsilon, \delta)$-indistinguishable distributions over $\Omega$ and $\widehat{P}$ be the distribution defined in eq. (12), then*

$$\left| \log \frac{\widehat{P}(\omega)}{Q(\omega)} \right| \leq 2\varepsilon - \log(1 - \delta') = \tilde{\varepsilon}$$

*for all $\omega \in \Omega$. Here, $\delta' = \frac{2\delta}{1 - e^{-\varepsilon}}$. That is to say, $\widehat{P}$ and $Q$ are $\tilde{\varepsilon}$-indistinguishable.*

**Error Metric.** Two of the normally used metrics for a private mechanism $M$ are the squared error (denoted by $\mathsf{err}_{MSE}$) and absolute error (denoted by $\mathsf{err}_{\ell_\infty}$), respectively:

$$\mathsf{err}_{MSE}(\mathcal{M}, A, n) := \max_{x \in \mathbb{R}^n} \mathbb{E}\left[ \frac{1}{n} \|\mathcal{M}(x) - Ax\|_2^2 \right]$$
$$\mathsf{err}_{\ell_\infty}(\mathcal{M}, A, n) := \max_{x \in \mathbb{R}^n} \mathbb{E}\left[ \|\mathcal{M}(x) - Ax\|_\infty \right]. \tag{13}$$

# B    HIGH-LEVEL OVERVIEW OF OUR TECHNIQUES

In this section, we briefly discuss some techniques and ideas that underpin our proof.

## B.1 UPPER BOUND ON MATRIX MECHANISM IN $\ell_p^p$ METRIC.

To find an upper bound on answering linear queries, we use the Gaussian mechanism that adds correlated noise based on a factorization of the query matrix $A$. Specifically, given a query matrix $A \in \mathbb{R}^{m \times n}$, we consider the additive noise mechanism $\mathcal{M}(x) = Ax + z$. For any factorization of $A = LR$ where $L \in \mathbb{R}^{m \times k}$ and $R \in \mathbb{R}^{k \times n}$, such a mechanism can be rewritten as $\mathcal{M}(x) = L(Rx + z')$ where $Lz'$ has the same distribution as $z$. Finally, we show that minimizing the $\ell_p^p$ error on such mechanism is equivalent to finding an "optimal" factorization of $A$, and the optimal error can be characterized by the generalized factorization norm $\gamma_{(p)}(A)$.

## B.2 LOWER BOUND ON OBLIVIOUS ADDITIVE NOISE APPROXIMATE DP MECHANISMS IN $\ell_p^p$ METRIC.

To prove a lower bound on mechanisms that add oblivious additive noise, we consider the convex sensitivity polytope $AB_1^n = \{Ay : y \in \mathbb{R}^n \text{ and } \|y\|_1 \leq 1\}$ of the query matrix $A \in \mathbb{R}^{m \times n}$. We use the following measurement introduced by Nikolov & Tang (2024):

$$\inf_{W \in \mathbb{R}^{m \times m}} \left\{ \sqrt{\operatorname{tr}_{p/2}(WW^\top)} : \exists v \in \mathbb{R}^m, AB_1^n + v \subseteq WB_2^m \right\} \tag{14}$$

to bound the minimum scale of the variance needed for the noise to achieve differential privacy.

Intuitively, if the measure of the sensitivity polytope $AB_1^n$ is larger (in terms of $\sqrt{\operatorname{tr}_{p/2}(WW^\top)}$), then it is harder to make two points in $AB_1^n$ indistinguishable. To formulate such intuition, we first establish a bridge between $\ell_p^p$ error and the covariance matrix $\Sigma_{\mathcal{M}}(x) \in \mathbb{R}^{m \times m}$ of the output distribution (Lemma 2.2). Next, a direct approach is to show that if an oblivious mechanism $M$ is $(\varepsilon, \delta)$-differentially private, then by a standard lower bound in Kasiviswanathan et al. (2010), the square root of the covariance matrix $\Sigma_{\mathcal{M}}(x)$ satisfies that

$$AB_1^n + v \subseteq \sqrt{\Sigma_{\mathcal{M}}(x)} B_2^m$$

for some $v \in \mathbb{R}^m$, which establishes a relationship between the infimum value in eq. (14) and the $\ell_p^p$ error. Finally, we apply Lemma 20 in Nikolov and Tang Nikolov & Tang (2024) to lower bound such infimum value by the general factorization norm $\gamma_{(p)}(A)$.

However, the lower bound in Kasiviswanathan et al. (2010) works in only high privacy regime. To get a lower bound for approximate DP algorithms for all $\varepsilon > 0$, we use the fact that output distributions of differentially private mechanisms under two adjacent datasets must be close under $\chi^2$-divergence. Consequently, we employ the $\chi^2$-divergence to set a lower bound on the minimum variance of the oblivious noise that must be introduced to achieve differential privacy. Since we do not assume the unbiasedness as inNikolov & Tang (2024), we have to consider the bias of the oblivious noise. However, we show that such a pipeline still works if the oblivious noise has a small bias. On the other hand, if the noise has a large enough bias, then one can show that the $\ell_p^p$ error is already large. Combined, we establish a lower bound that the $\frac{1}{p}$-root of the $\ell_p^p$ error is at least $\Omega((1 - \delta)\gamma_{(p)}(A)/\varepsilon)$ for any oblivious $(\varepsilon, \delta)$-DP mechanisms on any query matrix $A \in \mathbb{R}^{m \times n}$ with $\kappa(A) > 0$.

With the lower bounds on oblivious mechanisms, we use the standard reduction in Bhaskara et al. (2012) to obtain a worst case (in terms of the input $x \in \mathbb{R}^n$) lower bound for *general* $(\varepsilon, \delta)$-DP mechanisms that might be data-dependent.

## B.3 COMPARISON OF TECHNIQUES

As alluded to in the introduction, the focus of Nikolov and Tang Nikolov & Tang (2023) is the $\ell_p^2$ *instance optimality* of matrix mechanisms among *unbiased* mechanisms, while we instead focus on *the worst case $\ell_p^p$ optimality* of matrix mechanisms among *any* differentially private mechanisms. Here, we highlight key departures between this paper and previous works.

**Add/remove model v.s. Substitution model.** Our main departure compared to Nikolov & Tang (2024) lies in a different privacy notion, leading to different choice of sensitivity polytopes for the geometric arguments of lower bound. To elaborate, we recall a natural way to view linear query as a mean estimation task as suggested in Nikolov & Tang (2024):

Let $\mathcal{X}$ be the domain of data points, $X = (x_1, x_2, \cdots, x_k)$ be the dataset, and $Q : \mathcal{X} \to \mathbb{R}^m$ be the query workload. Then, answering $Q(X)$ is equivalent to doing mean estimation in the polytope $K_Q = \{Q(x) : x \in \mathcal{X}\}$. The authors in Nikolov & Tang (2024) considered a substitution DP model, where a data point can be replaced by another within the domain. If we further let $h \in \mathbb{R}^{\mathcal{X}}$ (where $|\mathcal{X}| = n$ in the notation of our paper) be the histogram of the dataset, then the substitution DP model corresponds to the neighboring notion where $\|h - h'\|_1 \le 2$ and $\sum_{x \in \mathcal{X}}(h_x - h'_x) = 0$ (i.e., bounded DP). This differs from the commonly used privacy notion for linear queries (as used in Bhaskara et al. (2012), Edmonds et al. (2020) etc.) where we only ask $\|h - h'\|_1 = O(1)$, representing a natural $\ell_1$ sensitivity, and is more like the add/remove DP model when translated back to mean estimation. [6]

Even if we restrict our discussion to the substitution DP model, the lowerbound for mean estimation from Nikolov and Tang would be in terms of $\Gamma_p(K_Q) := \inf\left\{\sqrt{\text{Tr}_{p/2}(AA^\top)} : K_Q \subseteq_{\leftrightarrow} AB_2^d\right\}$. In contrast, our lowerbound for linear queries is in terms of the factorization norm of the workload matrix $A$ associated with the query $Q$. These two are not comparable: imagine when the workload matrix consists of vectors that are very far from the origin, but are close to each other; then we have small $\Gamma_p(K_Q)$ but a substantially larger factorization norm of the workload matrix $A$. Therefore, a lower bound in terms of $\Gamma_p(K_Q)$ does not imply a lower bound in terms of the factorization norm $\gamma_{(p)}(A)$ as ours. Essentially, this is due to the fact that the most suitable choice for "sensitivity polytope" is different for mean estimation and linear queries.

**Technical comparisons.** Our departure in analysis and its complication compared to previous works stems from the fact that we do not assume unbiased mechanisms. Our different approach also means that our dependency on $\kappa$ appears only in the applicable range of the privacy parameter $\delta$, instead of showing up in the lower bound itself. We elaborate it next.

The lower bound of Nikolov & Tang (2024) combined techniques from Edmonds et al. (2020) for oblivious mechanisms with the classical results for unbiased estimators, i.e., they crucially rely on the estimator being unbiased. We first explain at a high level why they need the assumption of an unbiased mechanism.

Edmonds et al. (2020) showed that the variance of the one-dimensional private mechanism is lower bounded by the width of the underlying sensitivity polytope. For a data oblivious mechanism, as considered in Edmonds et al. (2020) in their first step, this almost immediately implies a lower bound. However, this might not always be true for an unbiased mechanism. In fact, since Edmonds et al. (2020) consider the $\ell_2$ error metric, they can assume without any loss of generality that the bias is $0$. This is not the case for $\ell_p^2$ error considered in Nikolov & Tang (2024) or $\ell_p$-error as considered in this paper.

This causes the departure of our proof technique from Edmonds et al. (2020) and Nikolov & Tang (2024) since we cannot assume that the bias is $0$ either by an assumption of unbiased mechanism or because of the choice of metric (i.e., $\ell_2$ error metric). We first show in Lemma 2.3 that the error would be large if the bias is large enough. So, the rest of our proof has to deal with the setting when the bias is small. In fact, using a case analysis based on the magnitude of bias is also helpful from another perspective: our lower bound depends only on $\gamma_{(p)}(\cdot)$ norm while the effect of minimum width of sensitivity polytope is reflected in the applicable range of $\delta$ when we consider any $\varepsilon > 0$ (including the low privacy regime). In general, the width of the sensitivity polytope can be $0$ as shown in Lemma 2.5, but as we show it is lower bounded by a constant for two important linear query matrices. Further, for general linear query matrices whose sensitivity polytope has a small minimum width, say $1/n$, our lower bound remains non-trivial, while Nikolov & Tang (2024) only provided a very weak lower bound (that is, dependent inversely on the dimension). We discuss it next.

For approximate differential privacy, Nikolov & Tang (2024) proved that any mechanism would have a large error either on the input $x$ or one of its neighbor $x'$. This is because they rely on a classical result from statistics, known as Hammersley-Chapman-Robins bound (Lemma A.14). To apply this bound, they need to prove that the $\chi^2$-divergence between the mechanism's output on two neighboring datasets is bounded. However, while this is true for $\varepsilon$-differential privacy, this is not true for $(\varepsilon, \delta)$-differential privacy because the support of the two mechanisms might differ. To ensure that

---

[6]We have also confirmed with the authors of Nikolov & Tang (2024) that if using add/remove DP as in our setting, a lot of results in their paper need to be modified. Consequently, there is no black-box reduction and it does not appear to simplify our current proof.

the two distributions have the same support, they use the general trick used in differential privacy (and, to our knowledge, first appeared in Kasiviswanathan & Smith (2014a)) and define a set as we defined in eq. (4). This set serves two purposes: (i) the $\chi^2$-divergence between both distributions is bounded, and (ii) the difference of the expectation of either of the two distributions restricted over the set is close to the original unrestricted distribution unless one of the two distributions has a large variance. We can now do the case analysis. In **case 1**, if the expectation of neither of the two distributions changes much, we can restrict our attention to the defined set. Otherwise, we are in **case 2**, where we just pick the distribution whose expectation changed by a lot and for which we are in the case where the variance is high. As a result, one can only prove that either $\mathcal{M}_{\text{unbiased}}(x)$ or $\mathcal{M}_{\text{unbiased}}(x')$ have a large error.

There is another price with this analysis. If we are in **case 2**, then their technique gives a lower bound on the variance that depends on the minimum width ($\kappa(\cdot)$ in our paper and $w_0(\cdot)$ in Nikolov & Tang (2024)). Due to Lemma 2.5, their result by itself is vacuous if the query matrix $A$ has linearly dependent rows. They alleviate this concern using the following trick: one can always find a random subspace, so the minimum width is at least the inverse of the dimension of the original sensitivity polytope under the projection onto that subspace. In other words, in **case 2**, we can only prove a lower bound with inverse dependence on the dimension. As a result, the lower bound is less useful as the dimension increases.

Since the reduction from the class of oblivious additive noise differentially private mechanism to the class of general differentially private mechanism follows from Bhaskar et al. Bhaskara et al. (2012), we only focus on the class of oblivious additive noise differentially private mechanism in the following exposition. Since the large bias case is easy to deal with (and already implies a lower bound on the error as shown in Lemma 2.3), we need to deal with the case when the mechanism has a small bias.

Dealing with the possibility of bias results in an extra term of $\mathbb{E}[\theta^\top z] \left| 1 - \frac{Q}{P} \right|$ in eq. (17), where $P$ and $Q$ are the distribution of the output of the mechanism on two neighboring datasets, $\theta \in \mathbb{R}^d$ and $z$ is the noise which is stochastically independent of the input. Since $\mathbb{E}[\theta^\top z]$ is not identically zero, this term finally results in an extra term of $\delta' \mathbb{E}[\theta^\top z]$ term. For a non-vacuous lower bound, this term has to be $o(w_{AB_1^n}(\theta))$ in all directions $\theta$. Using the fact that we are in the low bias case, we have an upper bound on $\mathbb{E}[\theta^\top z]$; this gives us an applicable range of $\delta'$, i.e., the value of $\delta$ for which the term $\mathbb{E}[\theta^\top z] \in o(w_{AB_1^n}(\theta))$, which in turn depends on the narrowest direction of $w_{AB_1^n}$. This narrowest direction is $\kappa(A)$ by definition.

## C  PROOF OF THEOREM 1.6

We use the observation made in Edmonds et al. (2020). Let $Q = \mathcal{Q}_{d,w}$ be the corresponding matrix of the $w$-way parity queries on the domain $\{-1, 1\}^d$. Then, $Q$ is the sub-matrix of a $2^d \times 2^d$ Hadamard matrix $H$ (see also Definition A.7) produced by selecting $\binom{d}{w}$ rows of $H$. We have the following lemma that gives the lower bound on $\kappa(Q)$. This allows us to set the range of $\delta$ and combined with the worst case lower bound Theorem 1.4 give an $(\varepsilon, \delta)$-DP lower bound for general mechanisms on answering parity queries.

**Lemma C.1** $\kappa(Q) \geq 2$.

**Proof:**  Let $\ell = \binom{d}{w}$ and let $q_1^\top, q_2^\top \cdots, q_\ell^\top$ be the rows of $Q$. We first note that since $Q$ contains $\ell$ rows of a Hardamard matrix, then each row of $Q$ is orthogonal to each other, and the $\ell_2$ norm of each row $q_i^\top$ is $2^{d/2}$ where $1 \leq i \leq \ell$. We recall that

$$\kappa(Q) := \min_{\theta^\top \theta = 1} \left( \max_{\|x\|_1 \leq 1} \theta^\top Q x - \min_{\|x\|_1 \leq 1} \theta^\top Q x \right). \tag{15}$$

First note that for any fixed unit vector $\theta = (\theta_1, \theta_2, \cdots, \theta_\ell)^\top \in \mathbb{R}^\ell$,

$$\|\theta^\top Q\|_2^2 = (\theta_1 q_1^\top + \cdots + \theta_\ell q_\ell^\top)(\theta_1 q_1 + \cdots + \theta_\ell q_\ell) = \sum_{i=1}^\ell \theta_i^2 q_i^\top q_i = 2^d \sum_{i=1}^\ell \theta_i^2 = 2^d,$$

where the second equality comes from that $q_i^\top q_j = 0$ for any $i \neq j$. Then, we choose $x_+ \in \mathbb{R}^{2^d}$ be the vector such that $\|x_+\|_1 = 1$ and $\theta^\top Q = cx_+$ for some scalar $c > 0$, and $x_- = -x_+$. Finally, observe that

$$\theta^\top Q x_+ - \theta^\top Q x_- = 2\theta^\top Q x_+ = 2\|\theta^\top Q\|_2 \cdot \|x_+\|_2 \geq 2\frac{\|\theta^\top Q\|_2}{\sqrt{2^d}} = 2,$$

where the inequality comes from the fact that $\|x_+\|_2 \geq \|x_1\|_1/\sqrt{2^d}$. Since for any $\theta$ we can always find such a pair of $x_+$ and $x_-$, then we have $\kappa(Q) \geq 2$ by eq. (15). $\quad\square$

**Lemma C.2** $\|Q\|_1 = \binom{d}{w} 2^{d/2}$.

**Proof:** As noted in Edmonds et al. (2020), the parity query matrix, $Q$, is the submatrix formed by choosing the appropriate $\binom{d}{w}$ rows of a $2^d \times 2^d$ unnormalized Hadamard matrix. In other words, $n = 2^d$ and $m = \binom{d}{w}$. Since the Hadamard matrix is orthogonal, the rows of $Q$ are linearly independent. Furthermore, there are $\binom{d}{w}$ singular values, all of which are $2^{d/2}$. Since $\|Q\|_1$ is just the sum of the singular values of $Q$, we have the result. $\quad\square$

Setting $n = 2^d$ and $m = \binom{d}{w}$ gives us the required bound and proof of Theorem 1.6.

## D  PROOF OF THE UPPER BOUND

We first state the theorem in its full generality for the ease of the readers.

**Theorem D.1** *Fix any $0 < \varepsilon, \delta < 1$ and $2 \leq p < \infty$. For any query matrix $A \in \mathbb{R}^{m \times n}$ and dataset $x \in \mathbb{R}^n$, there exists a factorization of $A = LR$ and a parameter $\sigma = \sigma(\varepsilon, \delta, R)$ such that the mechanism*

$$\mathcal{M}(x) := L(Rx + z)$$

*where each entry in $z$ is i.i.d sampled from $\mathcal{N}\left(0, \sigma^2\right)$ preserves $(\varepsilon, \delta)$-differential privacy. Moreover,*

$$\left(\mathbb{E}\left[\|\mathcal{M}(x) - Ax\|_p^p\right]\right)^{1/p} \leq 3\gamma_{(p)}(A) \cdot \sqrt{\frac{\log(1/\delta)\min\{p, \log(2m)\}}{2\varepsilon^2}}.$$

**Proof:** Let $\rho = \frac{\varepsilon}{3\sqrt{\log(1/\delta)}}$. Note that the factorization of query matrix $A$ is independent of $x$. Thus, the mechanism $\mathcal{M}(\cdot)$ can be considered as the post-processing of $Rx + z$. The $\ell_2$ sensitivity of $Rx$ can be bounded by

$$\|Rx - Rx'\|_2 \leq \max_{\|y\|_1=1} \|Ry\|_2 = \|R\|_{1 \to 2},$$

since $\|x - x'\|_1 \leq 1$ if $(x, x')$ is a pair of neighboring datasets. Then, let $\sigma^2 = \frac{\Delta^2}{2\rho^2}$ where $\Delta = \|R\|_{1 \to 2}$, by Lemma A.12, $Rx + z$ preserves $(\varepsilon, \delta)$-DP as well as $\mathcal{M}(x)$. For the utility part, we consider the Gaussian variable $z' = Lz$ and thus $z' \sim \mathcal{N}(0, \sigma^2 LL^\top)$. Then, the $\ell_p^p$ error can be formulated as

$$\left(\mathbb{E}\left[\|\mathcal{M}(x) - Ax\|_p^p\right]\right)^{1/p} = \left(\mathbb{E}\left[\|LRx + Lz - Ax\|_p^P\right]\right)^{1/p} = \left(\mathbb{E}\left[\|Ax + z' - Ax\|_p^P\right]\right)^{1/p}$$

$$= \left(\mathbb{E}\left[\|z'\|_p^p\right]\right)^{1/p} = \left(\sum_{i \in n} \mathbb{E}[|z_i'|^p]\right)^{1/p}$$

$$\leq \sqrt{\min\{p, \log(2m)\}} \left(\sum_{i \in n} (\mathsf{Var}[z_i'])^{\frac{p}{2}}\right)^{\frac{2}{p} \cdot \frac{1}{2}}$$

$$= \sqrt{\min\{p, \log(2m)\}} \cdot \sigma \sqrt{\mathsf{tr}_{p/2}(LL^\top)}$$

$$= \frac{1}{\sqrt{2}\rho} \sqrt{\min\{p, \log(2m)\}} \sqrt{\mathsf{tr}_{p/2}(LL^\top)} \|R\|_{1 \to 2}.$$

Letting $L$ and $R$ be the optimal factorization of $A$ yields the desired result. Here, the inequality comes from the standard bound on the $p$-th moment of the Gaussian variable (Proposition 2.5.2 in Vershynin Vershynin (2018)) and the union bound over all coordinates respectively. $\quad\square$

# E MISSING PROOFS FROM SECTION 2.1

Recall that
$$\Sigma_{\mathcal{M}}(x) = \mathbb{E}[(\mathcal{M}(x) - \mathbb{E}[\mathcal{M}(x)])(\mathcal{M}(x) - \mathbb{E}[\mathcal{M}(x)])^{\top}]$$
is the covariance matrix of $\mathcal{M}(x)$.

## E.1 PROOF OF LEMMA 2.2

The proof follows from the following set of derivation.

$$
\left(\mathbb{E}\left[\|\mathcal{M}(x) - Ax\|_p^p\right]\right)^{2/p} \geq \mathbb{E}\|\mathcal{M}(x) - Ax\|_p^2 = \mathbb{E}\left[\left(\sum_{i=1}^{d}(\mathcal{M}(x)_i - (Ax)_i)^{2 \cdot \frac{p}{2}}\right)^{\frac{2}{p}}\right]
$$

$$
\geq \left(\sum_{i=1}^{d}\left(\mathbb{E}\left[(\mathcal{M}(x)_i - (Ax)_i)^2\right]\right)^{\frac{p}{2}}\right)^{\frac{2}{p}}
$$

$$
= \left(\sum_{i=1}^{d}\left(\mathbb{E}\left[z_i^2\right]\right)^{\frac{p}{2}}\right)^{\frac{2}{p}} \geq \left(\sum_{i=1}^{d}\left(\mathbb{E}\left[z_i^2\right] - (\mathbb{E}z_i)^2\right)^{\frac{p}{2}}\right)^{\frac{2}{p}}
$$

$$
= \left(\sum_{i=1}^{d}\mathsf{Var}[\mathcal{M}(x)_i]^{\frac{p}{2}}\right)^{\frac{2}{p}} = \mathsf{tr}_{p/2}(\Sigma_{\mathcal{M}}(x)).
$$

## E.2 PROOF OF LEMMA 2.4

Note that $\theta^{\top}\mathcal{M}(x)$ preserves $(\varepsilon, \delta)$-differential privacy for any $\theta$ if $\mathcal{M}(\cdot)$ is $(\varepsilon, \delta)$-differentially private. Further, $w_{AB_1^n}(\theta) = \max_{v \in AB_1^n} \theta^{\top} v - \min_{v \in AB_1^n} \theta^{\top} v$. For any proposition $\mathcal{P}$, we let

$$
\mathbb{1}\{\mathcal{P}\} = \begin{cases} 1 & \text{if } \mathcal{P} \text{ is true} \\ 0 & \text{otherwise} \end{cases}.
$$

Let $S = S_{P,Q,2\varepsilon}$. By the definition of $\widehat{P}$, similar to Nikolov & Tang (2024), we have

$$
\begin{aligned}
|\mathbb{E}_{X \sim \widehat{P}}[X] - \mathbb{E}_{X \sim Q}[X]| &= \left|\int_{\mathbb{R}} x\widehat{P}(x) - \int_{\mathbb{R}} xQ(x)\right| \\
&= \left|\int_{\mathbb{R}\backslash S} \left(x\widehat{P}(x) - xQ(x)\right) + \int_{S} x\widehat{P}(x) - \int_{S} xQ(x)\right| \\
&= \left|\frac{Q(S)}{P(S)}\int_{S} xP(x) - \int_{S} xQ(x)\right| \\
&= \left|\frac{Q(S)}{P(S)}\mathbb{E}_{X \sim P}[X\mathbb{1}\{X \in S\}] - \mathbb{E}_{X \sim Q}[X\mathbb{1}\{X \in S\}]\right| \\
&\geq \underbrace{\left|\frac{Q(S)}{P(S)}\mathbb{E}_{X \sim P}[X] - \mathbb{E}_{X \sim Q}[X]\right|}_{S_1} \\
&\quad - \underbrace{\left|\frac{Q(S)}{P(S)}\mathbb{E}_{X \sim Q}[X\mathbb{1}\{X \notin S\}] - \mathbb{E}_{X \sim Q}[X\mathbb{1}\{X \notin S\}]\right|}_{S_2}.
\end{aligned}
\tag{16}
$$

We now bound the above two terms separately. Recall that $P$ and $Q$ are distributions of $M_\theta(x) = \theta^\top(Ax + z)$ and $M_\theta(x') = \theta^\top(Ax' + z)$ respectively, then

$$S_1 := \left|\frac{Q(S)}{P(S)}\mathbb{E}_{X\sim P}[X] - \mathbb{E}_{X\sim Q}[X]\right| \geq \left|\theta^\top A\left(\frac{Q(S)}{P(S)}x - x'\right)\right| - |\mathbb{E}[\theta^\top z]|\left|1 - \frac{Q(S)}{P(S)}\right|. \quad (17)$$

By Lemma A.16, $1 - \delta' \leq P(S) \leq 1$ and $1 - \delta' \leq Q(S) \leq 1$. Further, if we chose $\varepsilon$ and $\delta$ such that $\delta' = \frac{2\delta}{1-e^{-\varepsilon}} < \frac{1}{2}$, then

$$1 - \delta' \leq \frac{Q(S)}{P(S)} \leq \frac{1}{1-\delta'} \leq 1 + 2\delta'. \quad (18)$$

Now we consider the term

$$f(x, y) := \left|\theta^\top A\left(\frac{Q(S)}{P(S)}x - y\right)\right|.$$

We do a case analysis based on the ratio $\frac{Q(S)}{P(S)}$.

- **When $\frac{Q(S)}{P(S)} \geq 1$**, then

$$AB_1^n \subseteq K_{P,Q} := \left\{A\left(\frac{Q(S)}{P(S)}x - y\right) : \|x - y\|_1\right\}.$$

Therefore, there exists a pair of $(x_+, x'_+)$ with $(x_+ - y_+) \in B_1^n$ such that

$$f(x_+, y_+) = \left|\theta^\top A\left(\frac{Q(S)}{P(S)}x_+ - y_+\right)\right| = \frac{w_{AB_1^n}(\theta)}{2}.$$

- **When $\frac{Q(S)}{P(S)} < 1$**, then the set

$$K'_{P,Q} = \left\{\frac{P(S)}{Q(S)} \cdot A\left(\frac{Q(S)}{P(S)}x - y\right) : \|x - y\|_1\right\} = \left\{A\left(x - \frac{P(S)}{Q(S)}y\right) : \|x - y\|_1\right\}$$

contains $AB_1^n$. In this case, there also exists a pair of $(x_-, y_-)$ with $(x_- - y_-) \in B_1^n$ such that

$$\left|\theta^\top A\left(\frac{Q(S)}{P(S)}x_- - y_-\right)\right| = \frac{Q(S)}{P(S)}\frac{w_{AB_1^n}(\theta)}{2} \geq (1 - \delta')\frac{w_{AB_1^n}(\theta)}{2}.$$

Finally, we have that

$$\left|\frac{Q(S)}{P(S)}\mathbb{E}_{X\sim P}[X] - \mathbb{E}_{X\sim Q}[X]\right| \geq (1 - \delta') \cdot \frac{w_{AB_1^n}(\theta)}{2} - 2\delta'\mathbb{E}\theta^\top z. \quad (19)$$

Next, we try to bound the second term in eq. (16):

$$
\begin{aligned}
S_2 &= \left|\frac{Q(S)}{P(S)}\mathbb{E}_{X\sim P}[X\mathbb{1}\{X \notin S\}] - \mathbb{E}_{X\sim Q}[X\mathbb{1}\{X \notin S\}]\right| \\
&\leq \left|\frac{Q(S)}{P(S)}\mathbb{E}_{X\sim P}[(X - \mathbb{E}_P[X])\mathbb{1}\{X \notin S\}] - \mathbb{E}_{X\sim Q}[(X - \mathbb{E}_Q[X])\mathbb{1}\{X \notin S\}]\right| \\
&\quad + \left|\frac{Q(S)}{P(S)}\mathbb{E}_P[X] \cdot \mathbb{E}_P[\mathbb{1}\{X \notin S\}] - \mathbb{E}_Q[X] \cdot \mathbb{E}_Q[\mathbb{1}\{X \notin S\}]\right| \\
&\leq \underbrace{\left|\frac{Q(S)}{P(S)}\mathbb{E}_{X\sim P}[(X - \mathbb{E}_P[X])\mathbb{1}\{X \notin S\}]\right|}_{S_{21}} + \underbrace{|\mathbb{E}_{X\sim Q}[(X - \mathbb{E}_Q[X])\mathbb{1}\{X \notin S\}]|}_{S_{22}} \\
&\quad + \underbrace{\delta'\left|\frac{Q(S)}{P(S)}\mathbb{E}_{X\sim P}[X] - \mathbb{E}_{X\sim Q}[X]\right|}_{S_{23}}.
\end{aligned}
\quad (20)
$$

We bound each of these terms separately.

**Bounding $S_{21}$ and $S_{22}$** Using $\frac{Q(S)}{P(S)} \leq 1 + 2\delta'$, we have

$$
\begin{aligned}
S_{21} = \frac{Q(S)}{P(S)} &\left| \mathbb{E}_{X \sim P}[(X - \mathbb{E}_P[X]) \mathbb{1}\{X \notin S\}] \right| \leq (1 + 2\delta') \left| \mathbb{E}_{X \sim P}[(X - \mathbb{E}_P[X]) \mathbb{1}\{X \notin S\}] \right| \\
&\leq (1 + 2\delta') \sqrt{\mathbb{E}_{X \sim P}[(X - \mathbb{E}_P[X])^2] \mathbb{E}[\mathbb{1}\{X \notin S\}]} \\
&\leq (1 + 2\delta') \sqrt{\delta' \cdot \mathbb{E}_{X \sim P}[(X - \mathbb{E}_P[X])^2]} \leq (1 + 2\delta') \sqrt{\delta' \mathsf{Var}[\theta^\top \mathcal{M}(x)]}.
\end{aligned}
$$

Similarly, we see that

$$
S_{22} = \left| \mathbb{E}_{X \sim Q}[(X - \mathbb{E}_Q[X]) \mathbb{1}\{X \notin S\}] \right| \leq \sqrt{\delta' \mathsf{Var}[\theta^\top \mathcal{M}(x')]}.
$$

Therefore,

$$
\begin{aligned}
S_{21} + S_{22} = \frac{Q(S)}{P(S)} &\left| \mathbb{E}_{X \sim P}[(X - \mathbb{E}_P[X]) \mathbb{1}\{X \notin S\}] \right| + \left| \mathbb{E}_{X \sim Q}[(X - \mathbb{E}_Q[X]) \mathbb{1}\{X \notin S\}] \right| \\
&\leq \frac{Q(S)}{P(S)} \sqrt{\delta' \mathsf{Var}[\theta^\top \mathcal{M}(x)]} + \sqrt{\delta' \mathsf{Var}[\theta^\top \mathcal{M}(x')]} \qquad (21) \\
&\leq (1 + 2\delta') \sqrt{\delta' \mathsf{Var}[\theta^\top \mathcal{M}(x)]} + \sqrt{\delta' \mathsf{Var}[\theta^\top \mathcal{M}(x')]},
\end{aligned}
$$

where the last inequality is due to eq. (18).

**Bounding $S_{23}$** With a similar argument as in $S_1$, we have

$$
\begin{aligned}
S_{23} = \delta' &\left| \frac{Q(S)}{P(S)} \mathbb{E}_{X \sim P}[X] - \mathbb{E}_{X \sim Q}[X] \right| \leq \delta' \left| \theta^\top A \left( \frac{Q(S)}{P(S)} x - y \right) \right| + \delta' |\mathbb{E}[\theta^\top z]| \cdot \left| 1 - \frac{Q(S)}{P(S)} \right| \\
&\leq \delta' \left( \frac{w_{AB_1^n}(\theta)}{2} + 2\delta' \mathbb{E}\theta^\top z \right), \qquad (22)
\end{aligned}
$$

where the last inequality can be achieved under the same choice of $x$ and $y$ as in eq. (19).

Plugging the bound in eq. (21) and eq. (22) in to eq. (20), we get

$$
\begin{aligned}
S_2 &\leq S_{21} + S_{22} + S_{23} \\
&\leq \delta' \left( \frac{w_{AB_1^n}(\theta)}{2} + 2\delta' \mathbb{E}\theta^\top z \right) + (1 + 2\delta') \sqrt{\delta' \mathsf{Var}[\theta^\top \mathcal{M}(x)]} + \sqrt{\delta' \mathsf{Var}[\theta^\top \mathcal{M}(x')]} \qquad (23)
\end{aligned}
$$

Plugging eq. (17) and eq. (23) in eq. (16) and setting $(\varepsilon, \delta)$ such that

$$
\delta' \leq \min\{\frac{1}{16}, \frac{\varepsilon \cdot \kappa(A) \cdot n^{\frac{p-2}{2p}}}{12\gamma_{(p)}(A)}, \varepsilon^2\} \leq \frac{1}{2},
$$

for any fix $\theta$, we have that for every $x \in \mathbb{R}^n$, there exists an $x'$ such that $\|x - x'\|_1 \leq 1$ and

$$
\begin{aligned}
|\mathbb{E}_{X \sim \widehat{P}}[X] - \mathbb{E}_{X \sim Q}[X]| &\geq \left| \frac{Q(S)}{P(S)} \mathbb{E}_{X \sim P}[X] - \mathbb{E}_{X \sim Q}[X] \right| \\
&\qquad - \left| \frac{Q(S)}{P(S)} \mathbb{E}_{X \sim Q}[X \mathbb{1}\{X \notin S\}] - \mathbb{E}_{X \sim Q}[X \mathbb{1}\{X \notin S\}] \right| \\
&\geq (1 - 2\delta') \cdot \frac{w_{AB_1^n}(\theta)}{2} - 3\delta' \mathbb{E}\theta^\top z - (1 + 2\delta') \sqrt{\delta' \mathsf{Var}[\theta^\top \mathcal{M}(x)]} - \sqrt{\delta' \mathsf{Var}[\theta^\top \mathcal{M}(x')]} \\
&\geq \frac{(1 - 2\delta') \cdot w_{AB_1^n}(\theta)}{2} - \frac{\varepsilon \cdot \kappa(A)}{4\gamma_{(p)}(A)} \cdot \frac{\gamma_{(p)}(A)}{\varepsilon} - (2 + 2\delta') \sqrt{\delta' \mathsf{Var}[\theta^\top \mathcal{M}(x)]} \\
&\geq \left( \frac{1}{2} - 2\delta' \right) \cdot \frac{w_{AB_1^n}(\theta)}{2} - \frac{17}{8} \sqrt{\delta' \mathsf{Var}[\theta^\top \mathcal{M}(x)]},
\end{aligned}
$$

which completes the proof of Lemma 2.4. Here, the second last inequality comes from that $\mathcal{M}(\cdot)$ adds oblivious noise and thus $\mathsf{Var}[\theta^\top \mathcal{M}(x)] = \mathsf{Var}[\theta^\top \mathcal{M}(x')]$.

# F MISSING PROOFS IN SECTION 2.2

## F.1 PROOF OF THEOREM 2.8

The key step in proving Theorem 2.8 is applying Lemma 4.12 in Kasivishwanathan et al. Kasiviswanathan et al. (2010).

**Lemma F.1 (Kasiviswanathan et al. (2010))** *Suppose $X, Y$ are real-valued random variables with statistical difference at most $e^{1/2} - 1 + \delta$. Then, for all $a \in \mathbb{R}$, at least one of $\mathbb{E}[X^2]$ or $\mathbb{E}[(Y - a)^2]$ is $\Omega(a^2(1 - \delta)^2)$.*

The following lemma introduced $\varepsilon$ into the above lower bound.

**Lemma F.2 (Dwork and Roth Dwork & Roth (2014))** *Fix any $0 < \varepsilon \leq \frac{1}{2}$ and $\delta > 0$. Let $A(x) : \mathbb{R}^n \to \mathbb{R}$ be any randomized algorithm. If $A$ is $(\varepsilon, \delta)$-differentially private, then $A\left(\frac{1}{2\varepsilon}x\right)$ is $(1/2, \frac{e^{1/2} - 1}{e^{\varepsilon} - 1}\delta)$-DP.*

We first prove the following lemma based on Lemma F.1, which has also been claimed in Edmonds et al. (2020) (Lemma 26) but without a proof.

**Lemma F.3** *Let $w \in \mathbb{R}^n$ be any single query and $M(x) := w^{\top}x + z$ $(z \in \mathbb{R})$ be any data-independent mechanism that is $(\varepsilon, \delta)$-differentially private for $0 < \varepsilon \leq \frac{1}{2}$ and $0 \leq \delta \leq 1$, then $(\mathbb{E}[z^2])^{1/2} \geq \frac{1-\delta'}{C\varepsilon}\|w\|_{\infty}$ for some universal constant $C$. Here, $\delta' = \frac{e^{1/2} - 1}{e^{\varepsilon} - 1}\delta$.*

**Proof:** We consider the mechanism $M'(x) = 2\varepsilon M(\frac{1}{2\varepsilon}x) = w^{\top}x + 2\varepsilon z$. Let $\delta' = \frac{e^{1/2} - 1}{e^{\varepsilon} - 1}\delta$, then $M'$ is $(\frac{1}{2}, \delta')$-differentially private. Fix any pair of neighboring dataset $x$ and $x'$ such that $\|x - x'\|_1 \leq 1$. Let $X = M'(x)$ and $Y = M'(x')$ respectively. Then, it is easy to verify that

$$d_{TV}(X, Y) = \max_{S \subseteq \mathbb{R}} |\mathrm{Pr}[X \in S] - \mathrm{Pr}[Y \in S]| \leq e^{1/2} - 1 + \delta'$$

since $M'$ is $(\frac{1}{2}, \delta')$-differentially private and thus $\mathrm{Pr}[X \in S] \leq e^{1/2}\mathrm{Pr}[Y \in S] + \delta'$ for any $S$.

Next, let $X' = X - w^{\top}x = 2\varepsilon z$, $Y' = Y - w^{\top}x = 2\varepsilon z + w^{\top}(x' - x)$ and $a = w^{\top}(x' - x)$. Then $d_{TV}(X', Y') = e^{1/2} - 1 + \delta'$ and thus by Lemma F.1 (Lemma 4.12 in Kasiviswanathan et al. (2010)), we have

$$\mathbb{E}[z^2] = \frac{1}{4\varepsilon^2}\mathbb{E}[X'^2] = \frac{1}{4\varepsilon^2}\mathbb{E}[(Y' - a)^2] \geq \frac{(w^{\top}(x - x'))^2}{C\varepsilon^2}(1 - \delta')^2$$

for some universal constant $C$. Finally, choose the pair of neighboring datasets $x$ and $x'$ that maximizes $w^{\top}(x - x')$ completes the proof. $\square$

Now, we are ready to start the proof of Theorem 2.8.

**Proof:** (Of Theorem 2.8.) This proof can be considered as a complementary version of the proof in Nikolov & Tang (2024) and Edmonds et al. (2020) since they only focus on unbiased mean estimation or linear queries in $\ell_2^2$ metric. We recall the reader the notation $K \subseteq_{\leftrightarrow} L$ for $K, L \subset \mathbb{R}^m$. The notations means that there exists a $v \in \mathbb{R}^m$ such that $K + v \subseteq L$. We now restate Lemma 2.6 and give a proof here:

**Lemma F.4 (Restatement of Lemma 2.6 in (Nikolov & Tang, 2024))** *Let $\mathcal{M} : \mathbb{R}^n \to \mathbb{R}^m$ be any randomized mechanism and $A \in \mathbb{R}^{m \times n}$ be any matrix. If there exists some universal constant $C$ such that for any input $x \in \mathbb{R}^n$ and any $\theta \in \mathbb{R}^m$, it satisfies*

$$\mathrm{Var}[\theta^{\top}\mathcal{M}(x)] \geq \left(\frac{w_{\theta}(AB_1^n)}{C}\right)^2, \tag{24}$$

*then $AB_1^n \subseteq_{\leftrightarrow} C\sqrt{\Sigma_{\mathcal{M}}(x)}B_2^m$.*

**Proof:** Recall that

$$\Sigma_{\mathcal{M}}(x) = \mathbb{E}[(\mathcal{M}(x) - \mathbb{E}[\mathcal{M}(x)])(\mathcal{M}(x) - \mathbb{E}[\mathcal{M}(x)])^{\top}]$$

is the covariance matrix of $\mathcal{M}(x)$. Therefore, $\mathsf{Var}[\theta^{\top}\mathcal{M}(x)]$ can be written as

$$\sqrt{\mathsf{Var}[\theta^{\top}\mathcal{M}(x)]} = \sqrt{\theta^{\top}\Sigma_{\mathcal{M}}(x)\theta}.$$

Note that $\left\|\sqrt{\Sigma_{\mathcal{M}}(x)}\theta\right\|_2 = \left\|\sqrt{\Sigma_{\mathcal{M}}(x)}\theta\right\|_2 \cdot \|u\|_2$ for any $u \in B_2^m$. Therefore, by Cauchy-Schwarz inequality, we have that

$$\left\|\sqrt{\Sigma_{\mathcal{M}}(x)}\theta\right\|_2 \geq \max_{u \in B_2^m} \theta^{\top}\sqrt{\Sigma_{\mathcal{M}}(x)}u = \max_{v \in E} \theta^{\top}v$$

for $E = \sqrt{\Sigma_{\mathcal{M}}(x)}B_2^m$. In the above, the equality can be achieved if $u = \theta/\|\theta\|_2$.

Now, for any $v \in AB_1^n$, let $K_v = \{u - v : u \in AB_1^n\}$ be a convex body. Since $v \in AB_1^n$, for any $\theta \in \mathbb{R}^m$,

$$\max_{u \in K_v} \theta^{\top}u = \max_{w \in AB_1^n}\{\theta^{\top}w\} - \theta^{\top}v \geq 0.$$

This implies the following set of inequalities:

$$\max_{u \in K_v} \theta^{\top}u \leq \max_{u \in K_v} \theta^{\top}u + \max_{u \in K_v} -\theta^{\top}u = \max_{u \in K_v} \theta^{\top}u - \min_{u \in K_v} \theta^{\top}u = w_{\theta}(K_v) = w_{\theta}(AB_1^n).$$

Finally, the assumption in Lemma F.4 is equivalent to the following: for any $\theta \in \mathbb{R}^m$,

$$c\max_{w \in E} \theta^{\top}w \geq \max_{u \in K_v} \theta^{\top}u.$$

Since both $E = \sqrt{\Sigma_{\mathcal{M}}(x)}B_2^m$ and $K_v = AB_1^n - v$ (for some $v \in AB_1^n$) contain zero vector, we have $K_v \subseteq c\sqrt{\Sigma_{\mathcal{M}}(x)}B_2^m$. This completes the proof of Lemma F.4 (and Lemma 2.6). $\qquad\square$

In the view of Lemma F.4, we next show that for any direction $\theta \in \mathbb{R}^m$, $\sqrt{\mathsf{Var}[\theta^{\top}\mathcal{M}(x)]} \gtrsim \frac{1}{\varepsilon}w_{AB_1^n}(\theta)$ as long as $\mathcal{M}(x)$ is $(\varepsilon, \delta)$-differentially private.

**Lemma F.5** *Fix any $0 < \varepsilon < \frac{1}{2}$, $0 \leq \delta < 1$ and a query matrix $A \in \mathbb{R}^{m \times n}$. Let $C > 0$ be some universal constant. If $\mathcal{M}(\cdot)$ is an additive noise mechanism such that $\mathcal{M}(x) = Ax + z$ for any dataset $x \in \mathbb{R}^n$ and $\mathcal{M}(\cdot)$ preserves $(\varepsilon, \delta)$-differential privacy, then for any $x \in \mathbb{R}^n$ and any direction $\theta \in \mathbb{R}^m$, let $\delta' = \frac{e^{1/2}-1}{e^{\varepsilon}-1}\delta$, we have*

$$\sqrt{\mathsf{Var}[\theta^{\top}\mathcal{M}(x)]} \geq \frac{1-\delta'}{C\varepsilon}w_{AB_1^n}(\theta).$$

**Proof:** [Proof Of Lemma F.5] Given any vector $\theta \in \mathbb{R}^m$, recall that the width of a convex body $K \subseteq \mathbb{R}^m$ with respect to $\theta$ is defined as

$$w_K(\theta) := \max_{x \in K} \theta^{\top}x - \min_{x \in K} \theta^{\top}x$$

in Definition A.9. In this Lemma, we aim to show that if $\mathcal{M}(x)$ preserves $(\varepsilon, \delta)$-differential privacy, then the variance of the one-dimensional marginal of $\mathcal{M}(x)$ cannot be very small in terms of $w_{AB_1^n}(\theta)$. Unlike Nikolov & Tang (2024), since we focus on the additive noise mechanisms, we consider Lemma F.3 relating to the lower bound for such mechanisms.

Fix any $\theta \in \mathbb{R}^d$. We remark that we are trying the give the lower bound of the variance of one-dimensional marginal $\theta^{\top}\mathcal{M}(x)$, and $\theta^{\top}\mathcal{M}(x) = \theta^{\top}Ax + \theta^{\top}z$. By the post-processing property, $\theta^{\top}\mathcal{M}(x)$ also preserves $(\varepsilon, 0)$-differential privacy since $\mathcal{M}(x)$ is $\varepsilon$-differentially private. Thus, by Lemma F.3,

$$\mathbb{E}\left[(\theta^{\top}z)^2\right] = \mathsf{Var}[\theta^{\top}\mathcal{M}(x)] \gtrsim \frac{(1-\delta')^2}{\varepsilon^2}\|A\theta\|_{\infty}^2 \geq \frac{(1-\delta')^2}{\varepsilon^2}\max_{\|x-x'\|_1 \leq 1}|\theta^{\top}A(x-x')|^2. \quad (25)$$

Fix any $\theta \in \mathbb{R}^m$. We then show that for any $x$, there always exist a neighboring dataset $x'$ such that $|\theta^\top A(x - x')|$ can be lower bounded by $w_{AB_1^n}(\theta)/2$. This gives a lower bound of $\mathbb{E}\left[(\theta^\top z)^2\right]$. The construction closely follows Nikolov & Tang (2024) and we state the construction here for completeness.

Consider a mapping $f : \mathbb{R}^m \to \mathbb{R}^n$ from $AB_1^n$ to $B_1^n$ such that for any $v \in AB_1^n$, $v = Af(v)$. Given any $\theta \in \mathbb{R}^m$, let $w$ be the vector in $AB_1^n$ that maximizes $\theta^\top w$. Then, we can choose $x'_+$ such that $(x, x'_+)$ is a pair of neighboring datasets such that $x - x'_+ = f(w)$. In this case,

$$\theta^\top A(x - x'_+) = \theta^\top A f(w) = \theta^\top w = \max_{v \in AB_1^n} \theta^\top v.$$

Similarly, for any $x$ we can also find another pair of neighboring datasets $x$ and $x'_-$ such that

$$-\theta^\top A(x - x'_-) = \max_{v \in AB_1^n} -\theta^\top v = \min_{v \in AB_1^n} \theta^\top v.$$

Thus, we have

$$|\theta^\top A(x - x'_+)| + |\theta^\top A(x - x'_-)| = \left| \max_{v \in AB_1^n} \theta^\top v \right| + \left| \min_{v \in AB_1^n} \theta^\top v \right|$$

$$\geq \max_{v \in AB_1^n} \theta^\top v - \min_{v \in AB_1^n} \theta^\top v \geq w_{AB_1^n}(\theta).$$

In particular, this implies that, for any $\theta \in \mathbb{R}^m$ and any $x \in \mathbb{R}^n$, there exists an $x' \in \mathbb{R}^n$ neighboring to $x$ such that

$$|\theta^\top A(x - x')| \geq \frac{w_{AB_1^n}(\theta)}{2}, \quad \text{where} \quad w_K(\theta) := \max_{v \in K} v^\top \theta - \min_{v \in K} v^\top \theta. \tag{26}$$

Combining eq. (25) and eq. (26), we get

$$\mathsf{Var}[\theta^\top \mathcal{M}(x)] \gtrsim \frac{(1 - \delta')^2}{\varepsilon^2} \max_{\|x - x'\|_1 \leq 1} |\theta^\top A(x - x')|^2$$

$$\geq \frac{(1 - \delta')^2}{\varepsilon^2} |\theta^\top A(x - x')|^2 \geq (1 - \delta')^2 \left( \frac{w_{AB_1^n}(\theta)}{\varepsilon} \right)^2$$

for any $\theta \in \mathbb{R}^m$.

$\square$

We now proceed to complete the proof of Theorem 2.8. As a consequence of Lemma F.4 and Lemma F.5, by setting $W = C\varepsilon \sqrt{\Sigma_\mathcal{M}(x)}$, we see that for any $(\varepsilon, \delta)$-differentially private algorithm $\mathcal{M}$ and $p \geq 2$, there are

$$C\varepsilon \sqrt{\mathsf{tr}_{p/2}(\Sigma_\mathcal{M}(x))} \geq \inf_{W \in \mathbb{R}^{m \times m}} \left\{ \sqrt{\mathsf{tr}_{p/2}(WW^\top)} : AB_1^n \subseteq_\leftrightarrow WB_2^m \right\}$$

$$= \Lambda_p(A).$$

Then, by Lemma 2.7 and Lemma 2.2, we have

$$\left( \mathbb{E}\left[ \|\mathcal{M}(x) - Ax\|_p^p \right] \right)^{1/p} \geq \frac{1 - \delta'}{C\varepsilon} \cdot C\varepsilon \sqrt{\mathsf{tr}_{p/2}(\Sigma_M(X))}$$

$$\geq \frac{(1 - \delta')\Lambda_p(A)}{C\varepsilon} \geq \frac{(1 - \delta')\gamma_{(p)}(A)}{C\varepsilon},$$

which completes the proof of Theorem 2.8.

$\square$

## F.2 Proof of Lemma 2.5

Suppose $A$ has linearly independent rows, then for any non-zero $\widehat{\theta} \in \mathbb{R}^m$, $\widehat{\theta}^\top A$ must have at least one non-zero elements since $\widehat{\theta}^\top A$ would be linear combinations of rows of $A$. Let the non-zero element be $(\widehat{\theta}^\top A)_i > 0$. Then

$$\kappa(A) = \min_{\theta^\top \theta = 1} \left( \max_{x \in B_1^n} \theta^\top Ax - \min_{x \in B_1^n} \theta^\top Ax \right) \geq 2(\widehat{\theta}^\top A)_i > 0.$$

On the other hand, if $A$ has linearly dependent rows, then there will be a $\widehat{\theta} \in \mathbb{R}^n$ such that $\widehat{\theta}^\top A = 0$, and thus $\kappa(A)$. Intuitively, such $A$ maps $B_1^n$ to a lower dimension polytope.

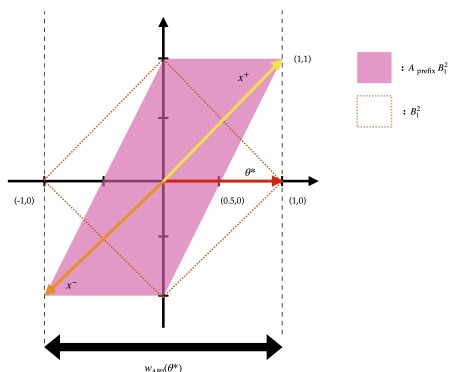

Figure 1: A geometric intuition of Lemma 2.14.

## G  MISSING PROOFS FROM SECTION 2.4

### G.1  PROOF OF LEMMA 2.14

**Proof:**  We first show that $\kappa(A_{\mathsf{prefix}}) \geq 2$. Fix any unit vector $\theta = (\theta_1, \cdots \theta_n)^\top \in \mathbb{R}^n$, we have

$$\max_{\|x\|_1 \leq 1} \theta^\top A_{\mathsf{prefix}} x = |\theta_1| + 2|\theta_2| + \cdots + n|\theta_n| = \sum_{i=1}^{n} i|\theta_i|. \tag{27}$$

We show the minimum value of eq. (27) by induction. We claim that for any $1 \leq i \leq n$, conditioned on $\sum_{j=1}^{i} \theta_j^2 = a$ for any $0 < a \leq 1$, the minimum value of $\sum_{j=1}^{i} j|\theta_j|$ is at least $\sqrt{c}$. Now, consider the new condition $\sum_{j=1}^{i+1} \theta_j^2 = c$ for some $0 < c \leq 1$. Let $a = c - \theta_{i+1}^2$, according to the assumption, we have that

$$\sum_{j=1}^{i} j|\theta_j| + (i+1)|\theta_{i+1}| \geq \sqrt{c - \theta_{i+1}^2} + (i+1)|\theta_{i+1}|.$$

Consider the function $f(y) = \sqrt{c - y^2} + (i+1)y$ for $0 < c \leq 1$ and $0 \leq y \leq \sqrt{c}$. We have $\frac{df}{dy} = \frac{-y}{\sqrt{c-y^2}} + i + 1$. Clearly for any $i \geq 1$, there exists a $0 < c_0 < \sqrt{c}$ such that $f(y)$ monotonically increasing in $(0, c_0)$ and monotonically decreasing in $(c_0, \sqrt{c})$. Thus, $f(y) \geq \min\{f(0), f(\sqrt{c})\} \geq \sqrt{c}$. That is,

$$\sum_{j=1}^{i} j|\theta_j| + (i+1)|\theta_{i+1}| \geq \sqrt{c}.$$

Since $c$ can be any value in $(0, 1]$ and $|\theta_1| = 1$ if $\theta_1^2 = 1$, by induction, for any unit vector $\theta \in \mathbb{R}^n$,

$$\max_{\|x\|_1 \leq 1} \theta^\top A_{\mathsf{prefix}} x = \sum_{i=1}^{n} i|\theta_i| \geq 1.$$

With a symmetric argument, we have that for the same vector $\theta$,

$$\min_{\|x\|_1 \leq 1} \theta^\top A_{\mathsf{prefix}} x = -\sum_{i=1}^{n} i|\theta_i| \leq -1,$$

Which implies that $\kappa(A_{\mathsf{prefix}}) \geq 2$. On the other hand, Let $\widehat{\theta} = \mathbf{e}_1 = (1, 0, \cdots, 0)^\top$, then it is easy to see that $w_{AB_1^n}(\widehat{\theta}) = 2$. Thus, $\kappa(A_{\mathsf{prefix}}) = 2$.  $\square$

Figure 1 also gives a geometric explanation of the most "narrow" width of $AB_1^n$ when $n = 2$. In the following diagram, $x^+ = (1, 1)^\top \in A_{\mathsf{prefix}} B_1^2$ and $x_- = (-1, -1)^\top \in A_{\mathsf{prefix}} B_1^2$.

## G.2 PROOF OF THEOREM 2.12

We show two factorizations of the counting matrix $A_{\mathsf{prefix}}$ that works across all $p$-norm for constant $p$. If we use the factorization of $A_{\mathsf{prefix}}$ as in Fichtenberger et al. (2023) or Henzinger & Upadhyay (2025). We here discuss Fichtenberger et al. (2023) for its simplicity. To recall their result, they construct matrices $L$ and $R$ such that $LR = A_{\mathsf{prefix}}$ and

$$L[i,j] = R[i,j] = \begin{cases} f(i-j) & i \geq j \\ 0 & i < j \end{cases}, \quad \text{where} \quad f(k) = \begin{cases} \left(1 - \frac{1}{2k}\right) f(k-1) & k \geq 1 \\ 1 & k = 0 \end{cases}$$

By noting that $f(k)$ is the Wallis' formula, we know that $f(k) \leq \frac{1}{\sqrt{\pi k}}$. This implies that

$$\|R\|_{1\to 2}^2 = \sum_{i=1}^{T} R[i,1]^2 = \sum_{i=1}^{n} f(i-1)^2 \leq 1 + \sum_{i=2}^{n} \frac{1}{\pi(i-1)} = O(\log(n))$$

Similarly,

$$\sqrt{\mathsf{tr}_{p/2}(LL^\top)} = \left(\sum_{i=1}^{n} (\|L[i,:]\|^2)^{p/2}\right)^{1/p} = O\left(\left(\sum_{i=1}^{n} \log^{p/2}(i)\right)^{1/p}\right) = O(n^{1/p}\sqrt{\log(n)}).$$

That is, $\gamma_{(p)}(A_{\mathsf{prefix}}) = O(n^{1/p}\log(n))$. Then, Theorem 2.12 follows using Theorem D.1.

