# OpenReview forum: "Optimality of Matrix Mechanism on $\ell_p^p$-metric"
_ICLR.cc/2025/Conference — ICLR 2025 Poster_

### Official Review · Reviewer_Gx7F · 2024-10-16

**Soundness:** 4
**Presentation:** 3
**Contribution:** 2
**Rating:** 6
**Confidence:** 4

**Summary:**

The paper studies worst-case lower bounds for answering linear queries under add-remove approximate differential privacy for an $\ell_p^p$ error metric. In contrast, past work derived instance-specific lower bounds for unbiased mechanisms under substitution approximation differential privacy in terms of $\ell_p^2$ metrics. Its main result is a lower bound  that is characterized by the factorization norm of the query matrix (Theorem 1.4). This lower bound is almost tight with an upper bound that is also provided using, as a long line of recent work has done, the matrix mechanism (Theorem D.1). The paper also specializes this result to the specific problems of prefix sum and parity by explicitly analyzing the problems' factorization norms (Theorem 1.5 and 1.6).

**Strengths:**

To the best of my knowledge, the claimed novelty in the paper is accurate, and it is nice to have a more complete lower bound picture for linear queries. The paper does a reasonable job of providing high-level explanations for some of its technical challenges and solutions. It is mostly easy to read with careful explanations.

**Weaknesses:**

To me, the paper's primary weakness is significance. Any one of the directions of novelty referenced above -- add/remove vs substitute, $\ell_p^p$ vs $\ell_p^2$, potentially biased vs unbiased -- feels pretty niche on its own, especially since we're only talking about lower bounds (as I think the authors would agree, the matrix mechanism upper bound is not a significant contribution here). As is, I don't think the paper convincingly argues why expanding known results in these directions is worth doing, beyond it just being something we could do. Since these are lower bounds, I want a stronger argument that these results somehow add "morally" to our understanding of private mechanisms -- and I'm not sure one exists (but, see questions below).

**Questions:**

Overall, I think expending more effort trying to explain why the directions of novelty add up to an "interesting" result could go a long way for this paper, but as is the qualitative improvement over known results feels weak. Some specifics:

1) The paper repeatedly emphasizes the distinction between add-remove and substitution, but this is described in a confusing way. Throughout, the database is presented as a vector in $\mathbb{R}^n$ -- what is $n$ here? Is it the size of the data universe, or the number of data points? Since neighboring databases both lie in $\mathbb{R}^n$ and the paper claims to study add-remove privacy -- where I'd expect neighboring databases to literally have # data points that differ by 1 -- I expect that $n$ is the size of the data universe and $x$ is a histogram, and Appendix B.3 seems to agree. But then, I don't know why it's more natural to allow neighboring databases to have $\|\|x-x'\|\|_1 \leq 1$, which is more like a strange "fractional" contribution model where a user can technically contribute $<1$ to many items. Edmonds-Nikolov-Ullman also appears to use the same neighboring notion as Nikolov-Tang, despite this paper's claim that it uses the same notion as this paper. Overall: I am confused by this bit.

2) Considering general $\ell_p$ norms typically feels to me like generalization for its own sake. The paper makes a few attempts to argue that $p \neq 2$ is more "natural", but the arguments are vague. The error metrics are clearly different, but what do they meaningfully add to our understanding? I think this paper would be stronger if the authors would try to make as cogent as possible a case that generalizing $p$ actually tells us something useful.

3) I have pretty much the same reaction to biased vs unbiased as I do to $p$ vs 2. I don't think anybody expected that biased mechanisms were the key to better worst-case error guarantees, so their inclusion again feels more like a technical wrinkle that necessitates different arguments than something that significantly adds to our understanding of privacy (and since it's lower bounds, I think adding to understanding matters more -- if there were new upper bounds, there would be a much simpler case that it's just a new useful thing).

---

> ### Author Response · Authors · 2024-11-18
>
> We thank the reviewer for the comments and suggestions.
>
> > About the privacy notion
>
> We would like to emphasize that the main distinction between substitution model and add/remove model is that they are related to different sensitivity polytope that are not directly comparable, and thus we need a new lower bound (see also the discussion in Appendix B.3). As the reviewer pointed out, $n$ is the size of data universe, and the number of data points is $\lVert x\rVert_1$. We define the privacy notion using $\lVert x - x'\rVert_1$ not to emphasize that we permit "fractional" contributions from multiple users (as the reviewer mentioned), but rather to just encompass the add/remove model. In this model, $\lVert x\rVert_1$ and $\lVert x'\rVert_1$ do not need to be equal because we allow the addition or removal of a single user. We do not intend to say that allowing fractional changes is more natural, we will make this point more clear.
>
> > "Edmonds-Nikolov-Ullman also appears to use the same neighboring notion as Nikolov-Tang"
>
> Edmonds-Nikolov-Ullman uses the same privacy notion as ours, as it does not require $\lVert x\rVert_1$ and $\lVert x'\rVert_1$ to be equal (see also page 26, specifically the discussion between Lemma 26 and Lemma 27 in the ENU paper).  Moreover, we have confirmed with the authors of NT24 in personal communication that they consider the substitution model, in which the number of users, $\lVert x\rVert_1$, is fixed.
>
> > "The error metrics are clearly different, but what do they meaningfully add to our understanding?"
>
> We believe studying $\ell_p$ error contributes to our understanding in the following aspects:
>
> 1. Studying the $\ell_p$ error for larger $p$ provides a characterization of the distribution of the error. To elaborate, let $v_i$ be the difference between the $i$-th coordinate of the output and the ground truth, then the error we studied is just $\mathbb{E}[|v_1|^p + \cdots + |v_n|^p]$. For any additive noise mechanism that adds noise from identical distribution, by the linearity of expectation, our lower bounds provides a lower bound for the $p$-th moment $\mathbb{E}[|v_i|^p]$ of the noise distribution. Therefore, obtaining the lower bound for various values of $p$ offers a more precise characterization of the "optimal" distribution, as it is well-known that the $p$-th moment for $p\geq 2$ provides substantial information about the distribution more than just $\mathbb{E}[|v_i|^2]$, which is implied by the squared error.
>
> 2. We note that our lower bound on $\ell_p$ error is tight, as it is accompanied with a matching upper bound. So it interpolates the curve of the "correct" magnitude of the error. For example, in the private prefix sum problem, we show that the tight error is $\tilde{\Theta}(n^{1/p})$, bridging the gap between the known bounds for squared error ($\tilde{\Theta}(\sqrt{n})$) and worst-case error ($\text{polylog}(n)$).
>
> 3. It serves as a unifying lens to understand several prior works including [ENU20], [ALNP24] and [NT24] since both [NT24] and [ALNP24] study the similar $\ell_p$ error.
>
> [ENU20] The Power of Factorization Mechanisms in Local and Central Differential Privacy
>
> [NT24] General Gaussian Noise Mechanisms and Their Optimality for Unbiased Mean Estimation
>
> [ALNP24] PLAN: Variance-Aware Private Mean Estimation
>
> > "I don't think anybody expected that biased mechanisms were the key to better worst-case error guarantees"
>
> In our experience, the best possible worst-case error is usually achieved by biased mechanism, so we are not sure if we understand the reviewer's comment correctly (please correct us if not). In the context of linear query, a line of works have been working on answering the sizes of all cuts privately in a graph, which can be considered as a special case of linear query whose workload matrix is the edge-adjacency matrix in terms of all $(S,T)$ cut. The optimal private algorithm on this problems that gives $\tilde{O}(\sqrt{mn})$ worst-case additive error is achieved by private mirror descent ([EKKL20]), which is a biased mechanism as it computes the private gradient by multiplicative Gaussian noise. Similarly, private multiplicative weights update (MWU) usually gives the optimal worst-case error for other well-known linear queries [Vad16] and it is also a biased mechanism because of its weighing of the hypothesize (and private) dataset. Apart from linear queries, exponential mechanism is usually considered as the optimal algorithm for many private problems, while discrete exponential mechanisms are typically biased (for example, consider the randomized response mechanism).
>
> [EKKL20] Differentially Private Release of Synthetic Graphs
>
> [Vad16] The Complexity of Differential Privacy

---

> > ### Comment · Reviewer_Gx7F · 2024-11-20
> > **Re: Official Comment by Authors**
> >
> > Overall, I'm sufficiently convinced to increase my score from weak reject to weak accept. I personally don't find this kind of paper very exciting, so I'm not going to fight for it, but I don't think it's an unreasonable fit at ICLR. Some details below:
> >
> > > We would like to emphasize that the main distinction between substitution model and add/remove model is that they are related to different sensitivity polytope...
> >
> > OK, this makes sense, I see how the $\|x-x'\|_1 \leq 1$ notation is a way to incorporate add-remove privacy. I'm more used to thinking of add-remove in terms of an $\ell_0$ and $\ell\infty$ constraint than an $\ell_1$ constraint, but it seems reasonable enough.
> >
> > > Edmonds-Nikolov-Ullman uses the same privacy notion as ours ...
> >
> > Their Section 2.2 (https://arxiv.org/pdf/1911.08339) says that adjacent databases have the same number $n$ of data points, and one is obtained by "replacing an element", which is substitution, not add-remove. However, the section you reference does appear to use the same $\|x-x'\|_1 \leq 1$ notion, so maybe their earlier definition is just mismatched with the rest of the paper.
> >
> > > We believe studying error contributes to our understanding in the following aspects ...
> >
> > I agree that the provided characterization fills a gap in the current understanding. However, my general feeling about studying  $\ell_p$ error for $2 < p < \infty$ is that it's often an answer to a question that nobody is really asking, in a (maybe narrow) practical sense. For me, this is compounded by the optimal matrix mechanism being a pretty impractical algorithm -- my understanding is that convergence to this optimum is only known in polytime for some highly restricted classes of polytopes (as described in NT24) and even then the polynomial is pretty bad. So, slightly more general lower bounds tight with an unrealistic algorithm don't feel very meaningful to me. I might be less interested in this kind of result than the median ICLR attendee, of course.
> >
> > I do think the added explanation given in this response would help the current submission's introduction.
> >
> > > In our experience, the best possible worst-case error is usually achieved by biased mechanism, so we are not sure if we understand the reviewer's comment correctly ...
> >
> > I had an incorrect notion of what "unbiased" means in mind. The provided examples are helpful -- I suggest putting at least the graph cut example in the submission.

---

> > > ### Author Response · Authors · 2024-11-21
> > >
> > > We thank the reviewer for raising the score and further sharing your constructive suggestions. We will definitely incorporate them to improve the paper.

---

### Official Review · Reviewer_BwNF · 2024-10-23

**Soundness:** 4
**Presentation:** 3
**Contribution:** 3
**Rating:** 8
**Confidence:** 3

**Summary:**

This paper studies the $\ell_p^p$-error of privately answering linear queries. The authors show a lower bound on the error in terms of a certain factorization norm ($\gamma_{(p)}(A)$), suggesting that the classical matrix factorization mechanism is optimal up to logarithmic factors. Based on this main result, they also provide an easy-to-use lower bound that depends on the Schatten-1 norm of the workload matrix. They then apply their results to two kinds of queries: prefix-sum and parity, and obtain nearly tight bounds for both tasks.

**Strengths:**

1. Linear query release is a fundamental problem in private data analysis. This article characterizes the error of this task with respect to $\ell_p^p$-metric, contributing to a deeper understanding of the role of differential privacy.
2. The writing is clear. Also, the technical details are well-organized and easy to follow.

**Weaknesses:**

1. It appears that the upper bound and lower bound only match in the high privacy regime. In the low privacy regime, where $\varepsilon = \Omega(1)$, the presented upper bound cannot match the lower bound.
2. When $p$ is not a constant, the proposed bounds may not be tight for certain regimes of the parameters. For example, when $n = p$, the upper bound in Theorem 1.5 is $O(\log^{1.5} (p))$ while the lower bound is $\Omega(\log (p))$.

However, I think both of the above are not serious issues.

**Questions:**

I do not understand what the workload matrix for parity queries is like. Can you provide a small example?

---

> ### Author Response · Authors · 2024-11-18
>
> We thank the reviewer for the comments and suggestions.
>
> >"I do not understand what the workload matrix for parity queries is like. "
>
> Depending on which parity queries are made, the workload matrix would consist of a subset of rows of a normalized $n \times n$ Hadamard matrix.

---

> > ### Comment · Reviewer_BwNF · 2024-11-26
> >
> > Thank you for your response.
> >
> > Could you incorporate a small toy example, e.g., $d = 2\text{ or }3$, in the revision?

---

> > > ### Author Response · Authors · 2024-11-27
> > >
> > > Thank you! We agree that including a toy example would improve the clarity of the workload matrix, and we will incorporate it into the revision.

---

### Official Review · Reviewer_AQvd · 2024-11-02

**Soundness:** 4
**Presentation:** 2
**Contribution:** 3
**Rating:** 6
**Confidence:** 2

**Summary:**

The paper investigates the asymptotic lower bounds of the expected $\ell_p$ errors of differentially private algorithms for answering linear queries. Specifically, given a collection of linear queries represented by a query matrix $A$, it presents an asymptotic error lower bound that any differentially private algorithm will incur, expressed in terms of the matrix $A$ and privacy parameters.

After establishing this lower bound, the paper demonstrates that the matrix mechanism can achieve it, up to logarithmic factors. Additionally, it explores two specific linear queries: the prefix sum and the parity queries, deriving their error lower bounds as special cases of the general lower bound obtained. It also presents specific matrix mechanisms that achieve asymptotic tight upper bounds (again, up to logarithmic factors).

**Strengths:**

1. The problem addressed is of fundamental importance.
2. The paper successfully derives asymptotic lower bounds for matrix mechanisms under $\ell_p$ error.
3. It demonstrates sophisticated applications of existing techniques.

**Weaknesses:**

1. The paper demonstrates that the matrix mechanism is asymptotically optimal, rather than instance optimal, for the error metric considered.
2. The writing is generally good, but there are parts that lack consistency.

**Questions:**

1. In Theorem 1.2, the distribution of $z$ does not incorporate the privacy parameter $\epsilon$. It seems that something is missing here.
2. In the next paragraph, change “also obtain a characterization of err() that is tight up to log factors” to “also obtain an err() that is tight up to log factors.”
3. Why is Theorem 1.3 included in the introduction? It appears disconnected from the surrounding discussion. For instance, do you use Theorem 1.3 to prove the lower bound of the prefix sum later in the paper?
4. Line 113: Change $v_1^p + \ldots + v_m^p$ to $|v_1|^p + \ldots + |v_m|^p$.
5. In line 113, why doesn’t the approach of Nikolov & Tang explain the additive noise mechanism? Is it because the additive noise mechanism might not be unbiased?
6. Is it common to encounter biased additive noise mechanisms? Would it be challenging to derive a lower bound for the biased case by reducing it to the unbiased one, using techniques similar to those presented in Lemma 2.3?
7. Could you elaborate a bit more on the term “symmetry” in line 128? What does it signify in this context? Also, in line 130, clarify “the mathematical object the sequence captures…”
8. Line 286: Change “the measure of the new distribution” to “the new distribution.”

---

> ### Author Response · Authors · 2024-11-18
>
> We thank the reviewer for the comments and suggestions.
>
> > "In Theorem 1.2, the distribution of $z$
>  does not incorporate the privacy parameter".
>
>  Thanks for catching this typo. We missed the privacy parameters here. The correct distribution should be $z\in \mathcal{N}(0, \lVert R \rVert_{1\rightarrow 2}^2 \cdot \sigma^2 \cdot \mathbb{I}_{k\times k})$ where $\sigma^2 = O(\frac{\log(1/\delta)}{\varepsilon^2})$.
>
> > "Why is Theorem 1.3 included in the introduction?"
>
> Theorem 1.3 is an easy-to-use lower bound. Since to apply Theorem 1.3, one only needs to compute the Schatten-1 norm (i.e., the sum of singular values) of the workload matrix, which is much easier than computing the $\gamma_{(p)}$ norm in the main theorem. Also, Theorem 1.3 is enough to derive the tight and explicit lower bound for prefix sum. We will make this point more clear.
>
> > "why doesn’t the approach of Nikolov \& Tang explain the additive noise mechanism?"
>
> For additive noise mechanisms, bounding the $\ell_p^p$ error is equivalent to giving a lower bound of $\mathbb{E}[|v_1|^p + \cdots +|v_n|^p]$ where $\{v_i\} (i\in [n])$ are noises.
> Suppose the algorithm adds identically distributed noise; by the linearity of expectation, we are able to characterize the $p$-th moment $\mathbb{E}[|v_i|^p]$ of the noise distribution. It is well-known that the $p$-th moment provides substantial information about the distribution. We will make this point more clear.
>
> > "Would it be challenging to derive a lower bound for the biased case by reducing it to the unbiased one, using techniques similar to those presented in Lemma 2.3?"
>
> We believe Lemma 2.3 is originally intended for making such reductions. We certainly do not rule out other possible ways for reduction, but we would like to emphasize that a black-box reduction from biased mechanism to unbiased mechanism in [NT24] does not give our result, as we use different privacy notion, which implies different choices of the sensitivity polytope (see also the discussions in Appendix B.3).
>
> > "Could you elaborate a bit more on the term “symmetry” in line 128? What does it signify in this context? Also, in line 130, clarify the mathematical object the sequence captures…"
>
> By symmetry, here we mean that all the powers and roots taken on the error vector is the same. That is, it is of the form
> $(|v_1|_p+ \cdots + |v_n|^p)$ or $(|v_1|_p+ \cdots + |v_n|^p)^{1/p}$ instead of one being $p$ and the other being $2$.
>
> When we consider $\ell_p$ norm, then it nicely interpolates between two popular norms studied in the literature, the Euclidean norm and max-norm. This is aligned with the Riesz's motivation to study the $\ell_p$ norm (also see footnote 1). For a given vector (or its distribution), computing its $\ell_p$ norm forms a sequence that converges to $\ell_\infty$ at one end and $\ell_2$ at the other. The behavior of this sequence is better understood than the metric considered in Nikolov-Tang, which also has the same limits at the either end, but behavior in the intermediate point is not very clear.

---

### Official Review · Reviewer_Qtjr · 2024-11-05

**Soundness:** 3
**Presentation:** 2
**Contribution:** 3
**Rating:** 8
**Confidence:** 3

**Summary:**

This paper studies the so-called "factorization mechanism" for answering linear queries privately. To recap, a workload of $n$ linear queries over a domain of size $m$ can be described by a matrix $A: [\pm 1]^{n\times m}$. The private query releasing algorithm asks to estimate $Ax$ while preventing the privacy of $x$ w.r.t. any adjacent $x'$ s.t. $\|x-x'\|_1 <= 1$.

The standard approach is to publish $Ax + N(0, sigma^2)$ where the Gaussian noise is calibrated to the sensitivity of $A$. The factorization mechanism tries to improve over this by writing $A = L R$ and publish $L ( Rx + N(0, (\sigma')^2))$ where $\sigma'$ is calibrated to the sensitivity of $R$ only.

There is a lot of work studying the power and limitations of the factorization mechanism, trying to argue it is the "optimal" mechanism for this task. Generally, the theme along this line of research is to fix an arbitrary algorithm $\mathcal{A}$ (not necessarily following the paradigm above), and let $\mathcal{B}$ be the optimal factorization mechanism (following the approach above with an optimal $L,R$). Then, argue that for some error metric $err$, one has $err(\mathcal{A}) \preceq err(\mathcal{B}))$ up to insignificant factors. The optimality of the factorization mechanism has been established for $\ell_2^2$-error metric, the $\ell^2_p$ metric (be a recent work [Nikolov-Tang'24].

The current paper is a close follow-up of [Nikolov-Tang'24]. It proved that the factorization mechanism is optimal under the $\ell^p_p$ metric, defined as
$$
err = \mathbb{E}[ \| \mathcal{A}(x) - Ax \|_p^p ]
$$
Namely, this is the $p$-th moment of the ``error vector''. Some motivations are provided, for one, as $p$ increases from $2$ to $\infty$, this error smoothly interpolates between the mean-squared error and the worst-case. Second, this paper argues that the error metric is more natural than $\ell_p^2$ norm considered in prior work.

The lower bound on error, implied by the proof, depends on a mysterious factorization norm and is hard to interpret. This paper gives some applications of their general lower bound to specific query families of interest. These include prefix-sum queries and sparse parity queries (or some might prefer to call it $k$-way marginals)

**Strengths:**

* I like the motivation of interpolating between $\ell_2$ and $\ell_\infty$ error via $\ell_p^p$.
* The introduction of the paper is largely clear and easy to follow (perhaps with some background knowledge).

**Weaknesses:**

* Although the gentle introduction is clear, once it goes into a more technical part, it becomes significantly harder for me to appreciate the discussion. I take Lines 254-255 "The main technical obstacle of Theorem 2.1 lies in ......" as the punchline of this paper. However, I have little idea what this means in terms of technical novelty. Given my limited time reading, I hope the authors can clarify some of my questions below.

* Some discussions in the introduction may be improved (see some of my questions below).

**Questions:**

* Line 111: you argued that $\ell_p^p$ is more natural. I somewhat agree with this point from a notational aesthetic point. However, can you comment on why the Nikolov-Tang paper was working with an "unnatural" measure of $\ell_p^2$? This might make your claim more convincing.

* Pargraphy beginning on Line 508: I struggled to understand why this paragraph means the Nikolov-Tang result is "instance optimal". Then I looked into their paper and found that their definition of instance optimality was somewhat nuanced, and your description was indeed accurate (but not super informative, IMO). I acknowledge it might be hard to convey the result of Nikolov-Tang concisely. But it might be worth rephrasing your conclusion a little bit.

* Technical question: I think the current Section 2 is quite dense, and you seem to be citing Nikolov-Tang heavily. Could you maybe try to distill some of your technical punchlines and pin down the exact manipulation where you differ from prior works? I would be ok if you defer a "complete proof sketch" to appendix or so.

---

> ### Author Response · Authors · 2024-11-18
>
> We thank the reviewer for the comments and suggestions.
>
> > "can you comment on why the Nikolov-Tang paper was working with an "unnatural" measure of $\ell_2^p$ norm?"
>
> From a technical perspective, [NT24] establishes a lower bound by first lower bounding the minimum variance of noise in the Euclidean geometry required to ensure privacy, where the $\ell_p^2$ lower bound is more straightforward to establish. We extend this analysis to the $\ell_p$ geometry. Through personal communication, the authors of [NT24] confirmed to us that they did not consider the $\ell_p^p$ metric considered in this paper (also see the footnote in page 2).
>
> > "I acknowledge it might be hard to convey the result of Nikolov-Tang concisely. But it might be worth rephrasing your conclusion a little bit."
>
> We would love to discuss the difference between instance optimality and worst-case optimality more formally in the conclusion part.
>
> > "Could you maybe try to distill some of your technical punchlines and pin down the exact manipulation where you differ from prior works?"
>
>
> Due to space constraints, we had to defer the detail discussion of the difference in our techniques and previous works to Appendix B.3. For the reviewer’s convenience, we also provide a high-level summary here. Please let us know if further clarification is needed.
>
> 1. A different privacy notion necessitates the choice of a different sensitivity polytope to exactly capture the geometry of error, and to relate to the factorization norm of the workload matrix itself.
>
> We first note that [NT24] does not imply a lower bound for the usual setting of linear queries due to different privacy notions. We have confirmed with the authors of Nikolov-Tang that if using the same privacy notion as in our setting, a lot of results in their paper need to be modified. To elaborate,
> [NT24] gives a bound for mean estimation in the substitution model of privacy. Using the histogram notation, let $h \in \mathbb{R}^{|\mathcal{X}|}$ (where $|\mathcal{X}| = n$ in the notation of our paper) be the histogram of the dataset, then the substitution DP model corresponds to the neighboring notion where $\lVert h - h'\rVert_1 \leq 2$ and $\sum_{x \in \mathcal{X}} (h_x - h’_x) = 0$.
>
> In contrast, our privacy notion for linear queries requires $\lVert h - h'\rVert_1 = O(1)$, representing a natural $\ell_1$ sensitivity, the same as [BDKT12], [ENU20] etc. Now answering linear queries, $Q$ over data $X = (x_1, x_2, \cdots, x_k)$, $Q(X)$ is equivalent to doing mean estimation in the polytope $K_Q = \{Q(x):x\in \mathcal{X}\}$. Even if we restrict our discussion to the substitution model, the lower bound for mean estimation from Nikolov and Tang would be in terms of $\Gamma_p(K_Q)$. In contrast, we get a bound in terms of the factorization norms of the workload matrix $A$ associated with the query $Q$. These two are not comparable:  imagine when the workload matrix consists of vectors that are very far from the origin but are close to each other; then we have small $\Gamma_p(K_Q)$ but a substantially larger factorization norm of the workload matrix $A$ (we are happy to include an example in our paper as well). Therefore, a lower bound in terms of $\Gamma_p(K_Q)$ does not imply a lower bound in terms of the factorization norm $\gamma_{(p)}(A)$ as ours.
> Essentially, this is due to the fact that the most suitable choice for "sensitivity polytope" is different for mean estimation and linear queries. This is especially important since we would like to give explicit lower bounds for explicit linear query families, and tight bounds are only possible with the right choice for "sensitivity polytope".
>
> 2. Removing unbiasedness assumption via the Bhaskar et al. reduction.
>
> Then, to remove the unbiasedness assumption, we chose a different path compared to [NT24] by first establishing a lower bound against data-oblivious mechanisms, where the ideas on handling the bias is summarized in Appendix B.3 (starting at line 887). We then follow the reduction of Bhaskar et al. to obtain lower bounds for general mechanisms.
>
> Finally, a side benefit of our approach is that the width of the sensitivity polytope along the narrowest direction, denoted by $\kappa$ in our paper and $w_0$ in [NT24], plays a different role in our lower bound than theirs. Unlike [NT24], our lower bound does not depend on $\kappa$ and it is crucial for our explicit lower bounds for the two classes of linear queries we discuss later.

---

> > ### Comment · Reviewer_Qtjr · 2024-11-26
> > **Thank yoyu for your response**
> >
> > Thank you for your response. That helped clarify my questions a lot. I have raised my score and encourage you to incorporate the discussions into the revision of your paper.

---

> > > ### Author Response · Authors · 2024-11-27
> > >
> > > Thank you! We appreciate your helpful comments and will certainly incorporate them into the next version of our paper.

---

### Meta-Review · Area_Chair_NYxX · 2024-12-14

**Metareview:**

## Summary of Contributions

This paper studies linear queries problem under differential privacy. The problem can be thought of as outputting an estimate of $Ax$ where $A$ is the workload matrix and $x$ is the histogram of the input dataset. Traditionally, most works have studied this under the $\ell_2^2$ and $\ell_\infty$ errors. Recently, Nikolov and Tang (ITCS, 2024) studied the problem under $\ell_p^2$ error for any $p \geq 2$ and show that the so-called matrix mechanism (aka factorization mechanism) yields a nearly optimal error among all unbiased mechanisms. This paper shows a similar result but for an arguably more natural $\ell_p^p$ error and without the unbiased assumption.

## Strengths

- Linear queries are one of the most important and well-studied setting in private data analysis. This paper advances our understanding on the problem.

- $\ell_p^p$ errors are natural extensions of $\ell_2^2$ and $\ell_\infty$ errors, and are more natural than the $\ell_p^2$ error studied in (Nikolov & Tang, 2024).

- The proofs are sophisticated and require clever use of existing techniques.

## Weaknesses

- It is not entirely clear how important $\ell_p^p$ errors are since, for most previous applications, $\ell_2^2$ and $\ell_\infty$ seem sufficient.

- The lower bounds in this paper are for worst-case errors whereas (Nikolov & Tang, 2024) gives an "instance-optimality" result, which is a stronger type of lower bound. (This is not a strong point since all work prior to (Nikolov & Tang, 2024) also studied worst-case errors.)

## Recommendation

Given the importance of linear queries and that this paper gets nearly-tight bounds for $\ell_p^p$ errors, we recommend acceptance.

**Additional Comments On Reviewer Discussion:**

The reviewers asked for a few clarifications, including the adjacency notions used in the paper (add-remove DP) which is different compared to the one used in Nikolov-Tang (substitution DP) and why the techniques from the former do not apply to $\ell_p^p$. The authors provide satisfactory answers that clarify these confusions.

---

### Decision · Program_Chairs · 2025-01-22

Accept (Poster)